# FastTracker: Real-Time and Accurate Visual Tracking

## Abstract

Conventional multi-object tracking (MOT) systems are predominantly designed for pedestrian tracking and often exhibit limited generalization to other object categories. This paper presents a generalized tracking framework capable of handling multiple object types, with a particular emphasis on vehicle tracking in complex traffic scenes. The proposed method incorporates two key components: (i) an occlusion-aware re-identification mechanism that enhances identity preservation for heavily occluded objects, and (ii) a road-structure-aware tracklet refinement strategy that utilizes semantic scene priors—such as lane directions, crosswalks, and road boundaries—to improve trajectory continuity and accuracy. In addition, we introduce a new benchmark dataset comprising diverse vehicle classes with frame-level tracking annotations, specifically curated to support evaluation of vehicle-focused tracking methods. Extensive experimental results demonstrate that the proposed approach achieves robust performance on both the newly introduced dataset and several public benchmarks, highlighting its effectiveness in general-purpose object tracking. While our framework is designed for generalized multi-class tracking, it also achieves strong performance on conventional pedestrian benchmarks, with HOTA scores of 66.4 on MOT17 and 65.7 on MOT20 test sets.

## 1 Introduction

Multi-object tracking (MOT) is fundamental in applications such as surveillance and autonomous driving, yet remains challenging due to target similarity, frequent occlusions, and dynamic scene changes (Milan et al., 2016). The dominant *tracking-by-detection* paradigm (Babaee et al., 2017) addresses MOT by first detecting objects in each frame and then associating them with tracklets, often via similarity-based matching and algorithms like the Hungarian method.

While effective, most existing frameworks are benchmarked on single-class scenarios—primarily pedestrian tracking—where detectors are tuned for a specific category. This narrow focus inflates performance and overlooks the complexity of real-world deployments, where multiple object types coexist. Extending detection to multi-class settings typically reduces accuracy, leading to degraded tracking performance (Hao et al., 2024), underscoring the need for general solutions.

Multi-class tracking must account for varying reliability across detection confidences: high-confidence boxes usually correspond to true positives, while low-confidence ones are prone to false alarms. ByteTrack (Zhang et al., 2022) addresses this by first matching high-confidence detections, then selectively associating low-confidence candidates with unmatched tracks. Building on this idea, our framework adopts a similar two-stage process with tailored thresholds—first linking high-confidence detections to active tracklets using a

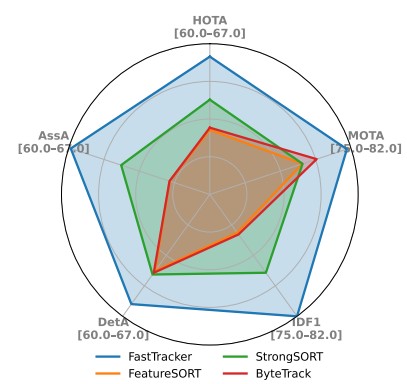

Figure 1: Comparison of multi-object tracking performance using five key metrics: HOTA, MOTA, IDF1, DetA, and AssA on MOT17 test set. Each axis is within the specified range, highlighting trade-offs between each metric.

relaxed criterion to maximize recall, and then applying a stricter constraint for low-confidence matches to preserve reliability. Unlike many trackers that depend on CNN-based appearance features (Yu et al., 2016; Hashempoor et al., 2024), our method emphasizes motion cues such as spatial proximity, bounding box geometry, and velocity consistency, making it more efficient for real-time use. To further improve robustness in challenging scenarios, we integrate an occlusion-handling module that re-identifies temporarily lost objects and exploit scene-level context (e.g., road layouts, traffic flow, crosswalks) to guide trajectory refinement. Together, these strategies significantly enhance tracking reliability while avoiding the overhead of deep ReID networks. The performance trends of our tracker compared to existing methods, measured across multiple metrics, are in Figure 1.

To handle occlusions without relying on appearance-based re-identification, we exploit confidence history and spatial interactions to infer occlusion events, and introduce a coverage metric with geometric overlap heuristics to identify hidden targets. When a target is marked as occluded, its Kalman updates are moderated to avoid unrealistic drift, enabling stable re-identification after reappearance (see Fig. 2). In addition, we incorporate scene-level structural priors by modeling roads and crosswalks as polygonal regions with dominant entry–exit edges, enforcing motion consistency along plausible directions (e.g., one-way flows). Trajectories that deviate significantly from these priors are corrected back into the valid motion corridor, which reduces identity switches caused by occlusions or detector noise. Together, these two components improve robustness in crowded, structured urban scenes while remaining lightweight. Finally, we also revise track initialization and deletion policies, which are typically overlooked in existing trackers, to reduce spurious identities and improve recovery of true targets.

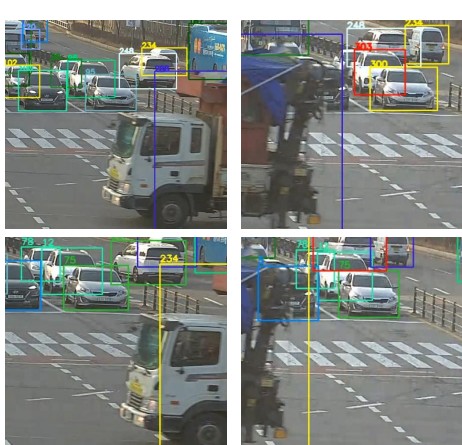

Figure 2: Tracking results without (top) and with (bottom) occlusion handling. Our method preserves vehicle IDs after severe occlusion, while the baseline shows frequent ID switches.

While our framework addresses occlusion and scene-structure challenges, existing benchmarks still limit progress by focusing mainly on pedestrian tracking (e.g., MOT17 (Dendorfer et al., 2020a), MOT20 (Dendorfer et al., 2020b)) or offering vehicle datasets with restricted object classes and simplified settings. To fill this gap, we introduce a new CCTV-based benchmark featuring diverse object categories—cars, trucks, buses, motorcycles, and pedestrians—in complex urban environments with occlusions, dense intersections, and multi-directional flows. This dataset provides a realistic testbed for multi-class tracking and encourages algorithms that generalize scenarios.

The main contributions of this work are: (i) a robust multi-class MOT framework that generalizes beyond pedestrian tracking and performs effectively on diverse vehicle classes in complex urban scenes; (ii) a lightweight occlusion-handling module that avoids reliance on appearance features or deep ReID networks by using spatial cues and geometric coverage, complemented by occlusion-aware initialization and deletion policies to reduce spurious identities and improve recovery of true targets; (iii) environment-aware constraints based on road geometry and scene semantics to enforce plausible motion and improve re-identification; (iv) a new CCTV-based benchmark dataset covering pedestrians and multiple vehicle categories in challenging scenarios such as occlusions and multi-directional traffic; and (v) state-of-the-art performance on MOT16, MOT17, MOT20, DanceTrack, and BDD100K, with HOTA scores of 66.0, 66.4, 65.7, 65.9, and an mMOTA of 43.8 on BDD100K, surpassing most existing trackers, outperforming most of computationally expensive deep visual tracking baselines.

## 2 RELATED WORK

In the tracking-by-detection paradigm, detections from deep networks (Sun et al., 2021) are linked across frames using geometric or appearance cues. While appearance features (e.g., FeatureSORT, DeepSORT (Wojke et al., 2017), or SSM-based extensions (Hashempoorikderi & Choi, 2024; Wang et al., 2024a)) improve robustness, they require additional Re-ID networks that increase memory and runtime cost, limiting their practicality in real-time applications. Lightweight alternatives such as SORT (Bewley et al., 2016), OC-SORT (Cao et al., 2023), and BoT-SORT (Aharon et al., 2022) demonstrate that motion-only designs can achieve strong performance while remaining efficient, motivating further exploration of geometry-driven approaches.

Occlusion remains a central challenge in MOT. ByteTrack (Zhang et al., 2022) partially addresses this by associating low-confidence detections, but its performance drops in crowded scenes. CNN-based Re-ID methods (Zhang et al., 2023) improve recovery but incur high computational cost. Geometry-driven methods such as PD-SORT (Wang et al., 2025) and SparseTrack (Liu et al., 2025) attempt to use pseudo-depth cues but often fail under non-ideal perspectives. In parallel, contextual information has been used to constrain tracking, e.g., predefined ROIs (Sani & Anand, 2024; Huang et al., 2023) or multi-camera entry–exit regions (Chai et al., 2024). However, these do not correct drifts within a single camera view. Our work instead combines occlusion handling without appearance models and scene-level geometry to adapt tracklets in complex urban layouts efficiently.

Several benchmarks support multi-object tracking but have limited coverage of urban CCTV scenarios. Waymo (Sun et al., 2020a) and KITTI (Geiger et al., 2012) target autonomous driving with restricted perspectives, while LMOT (Wang et al., 2024b) and VETRA (Hellekes et al., 2024) focus on nighttime or aerial views, respectively. In contrast, our benchmark emphasizes multi-class urban surveillance with diverse categories, occlusions, and bidirectional traffic. Here we highlight the most relevant works and key references; a more extensive discussion of related methods is in Appendix A.

## 3 BACKGROUND

In tracking-by-detection frameworks such as ByteTrack, data association is performed in two stages to balance recall and precision. Let $\mathcal{T}$ denote the set of tracklets at the current frame, and let the detector provide two groups of detections: $\mathcal{D}_{\text{high}}$ for high-confidence detections (above threshold $\tau_{\text{high}}$) and $\mathcal{D}_{\text{low}}$ for low-confidence detections (between $\tau_{\text{low}}$ and $\tau_{\text{high}}$). In the first stage, $\mathcal{D}_{\text{high}}$ is matched with $\mathcal{T}$ using IoU-based similarity, yielding matched pairs $(\mathcal{T}_1, \mathcal{D}_1)$. Unmatched tracklets and detections are retained as $\mathcal{T}_{\text{remain}}$ and $\mathcal{D}_{\text{remain}}$, respectively, for further matching in the second stage and initializing new tracklets. In the second stage, the unmatched tracklets $\mathcal{T}_{\text{remain}}$ are further matched with $\mathcal{D}_{\text{low}}$, producing additional associations $(\mathcal{T}_2, \mathcal{D}_2)$. This cascaded design ensures that reliable high-confidence detections are prioritized, while low-confidence detections are selectively used to recover missed targets without introducing excessive false positives.

## 4 APPROACH AND BENCHMARK

Building on the two-stage matching scheme described in Background 3, our framework adopts high/low-confidence two stages association design as its foundation. While effective, this strategy alone is insufficient under severe occlusion and dense traffic, where ambiguities often cause identity switches or drift. Moreover, naive initialization and deletion policies in baselines like ByteTrack can further degrade performance. To address these shortcomings, our framework augments the baseline with three key components: (1) explicit occlusion detection metrics based on spatial overlap, enabling occluded targets to be identified and maintained even without detections; (2) environment-aware constraints—such as road directionality, street layout, and pedestrian zones—that refine trajectories and suppress implausible motions; and (3) revised initialization and deletion policies that limit spurious identities while improving recovery of true targets. Together, these additions significantly enhance robustness in complex, multi-class urban scenes. The complete FastTrack algorithm is summarized in Figure 3. The full tracking algorithm is in Algorithm 1 in Appendix.

**Motion Prediction.**  For each tracklet $t \in \mathcal{T}$, we estimate its future state $\hat{s}_t$ using a class-aware NSA Kalman filter, $\text{KalmanPredict}(t)$. The motion model parameters are selected based on the object class: vehicles such as cars or motorcycles are allowed higher velocities and acceleration

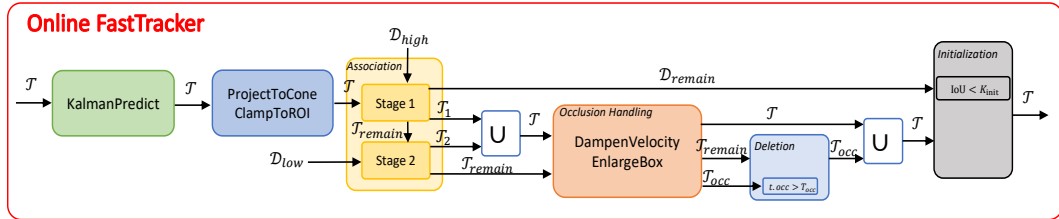

Figure 3: Overall pipeline of Online FastTracker. Kalman prediction is refined with region constraints (`ProjectToCone`, `ClampToROI`), followed by two-stage association using high- and low-confidence detections. Occlusion handling (`DampenVelocity`, `EnlargeBox`), deletion, and initialization ensure stable and consistent tracklet management.

bounds, whereas pedestrians are modeled with smoother and slower dynamics. This enables more realistic trajectory predictions, especially in cases of temporary occlusion or detection dropout.

**Direction and ROI Constraint.** To prevent tracklets from drifting into implausible directions, we retrieve a region $R$ around the predicted state $\hat{s}_t$ via a region lookup on the environment map $\mathcal{M}$. Each region encodes semantic layout (e.g., drivable roads, pedestrian paths) and is associated with a motion cone that specifies the allowed directionality. The cone's opening angle $\theta$ is derived from the polygonal structure of $R$: given an entrance edge and an exit edge (defined by prior scene-flow knowledge), we construct the two crossing diagonals that connect opposite endpoints of the entrance and exit edges. Let the entrance edge endpoints be $E_1, E_2$ and the exit edge endpoints be $O_1, O_2$, with diagonals $\overrightarrow{E_1O_2}$ and $\overrightarrow{E_2O_1}$, then

$$\theta = \arccos\left(\frac{(O_2 - E_1)\cdot(O_1 - E_2)}{\|O_2 - E_1\|\,\|O_1 - E_2\|}\right). \quad (1)$$

For each active tracklet $t$, we also compute its instantaneous motion direction $\phi$. Let the track center positions at frames $k$ and $k - N$ be $p_k = (x_k, y_k)$ and $p_{k-N} = (x_{k-N}, y_{k-N})$. The displacement $\Delta p = p_k - p_{k-N} = (\Delta x, \Delta y)$ gives $\phi/2 = \arctan(\Delta y, \Delta x)$. The function $\mathrm{ProjectToCone}(\hat{s}_t, \mathrm{cone}(R))$ then projects the predicted position into the direction-constrained cone (i.e., enforces $\phi \in [-\theta/2, \theta/2]$ around the region's dominant flow),

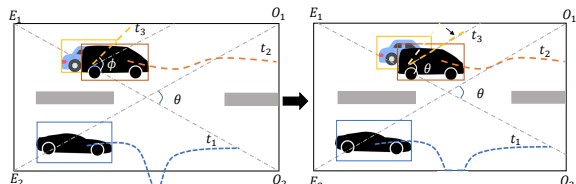

Figure 4: Illustration of `ClampToROI` and `ProjectToCone`. Tracklet $t_1$ drifts outside the region and is clamped back inside. Tracklet $t_3$ is misled by occlusion from $t_2$, causing its motion angle $\phi$ to exceed the allowed threshold $\theta$; we project it back within the motion cone.

and $\mathrm{ClampToROI}(\hat{s}_t, R)$ keeps the state inside the region. We illustrate these concepts in Figure 4 for a one-way road; the same construction extends to two-way roads and crosswalks.

**Association.** In the first stage, high-confidence detections $\mathcal{D}_{\mathrm{high}}$ are matched with active tracklets $\mathcal{T}$ using IoU-based association, yielding matches $(\mathcal{T}_1, \mathcal{D}_1)$. Unmatched tracklets and low score detections, $\mathcal{T}_{\mathrm{remain}}$ and $\mathcal{D}_{\mathrm{low}}$, are then passed to the second stage, where $\mathcal{T}_{\mathrm{remain}}$ is matched with low-confidence detections $\mathcal{D}_{\mathrm{low}}$ to recover hard cases $\mathcal{T}_2$ and we exclude them from $\mathcal{T}_{\mathrm{remain}}$. Finally, the updated active set becomes $\mathcal{T} = \mathcal{T}_1 \cup \mathcal{T}_2$.

**Occlusion Handling.** Unmatched tracklets $\mathcal{T}_{\mathrm{remain}}$ are checked for occlusion by measuring spatial overlap with active tracklets $\mathcal{T}$. If the overlap score $\mathrm{IoU}(t, t')$ between tracklets exceeds a threshold $\mathrm{IoU}_{\min}$, the tracklet $t$ is marked as occluded and added to $\mathcal{T}_{\mathrm{occ}}$. For each occluded tracklet, we mark the tracklet as occluded and apply two corrective operators to stabilize its state during disappearance: (i) velocity damping and (ii) bounding-box enlargement, shown in Figure 5. For velocity damping, we modify the Kalman state of a tracklet $t$ with position $(x_k, y_k)$ and velocity $(\dot{x}_k, \dot{y}_k)$ at frame $k$. During occlusion, the operator $\mathrm{DampenVelocity}(t)$ resets the velocity magnitude using a damping factor $\gamma_{\mathrm{velo}} \in (0, 1)$ and offset $\delta_v$, while the position is softly reset toward its last non-occluded location with offset $\delta_p$:

$$(\dot{x}_k, \dot{y}_k) \leftarrow \gamma_{\mathrm{velo}} \cdot (\dot{x}_{k-\delta_v}, \dot{y}_{k-\delta_v}) \quad \text{and} \quad (x_k, y_k) \leftarrow (x_{k-\delta_p}, y_{k-\delta_p}). \quad (2)$$

This reset constrains unrealistic forward propagation, while subsequent Kalman updates continue from the corrected state. For bounding-box enlargement, to improve re-identification after reappearance, the bounding box dimensions are slightly enlarged depending on the object class. If $(w, h)$ denote the width and height, then the operator `EnlargeBox(t)` applies:

$$(w, h) \leftarrow \beta_{\text{enlarge}} \cdot (w, h). \qquad (3)$$

This adjustment increases tolerance for detector noise and partial visibility when the occluded object becomes visible again. Occluded tracklets $\mathcal{T}_{\text{occ}}$ are not excluded from the matching process; instead, they are protected from deletion by applying a dedicated occlusion threshold, allowing them to persist until potential re-identification. The effectiveness of velocity damping and bounding-box enlargement is in Figure 6.

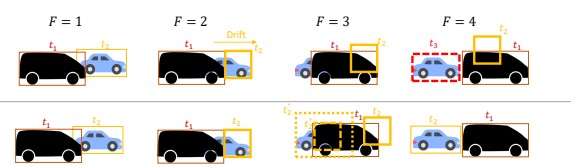

Figure 5: Illustration of `DampenVelocity` and `EnlargeBox` over four frames ($F = 1 - 4$). Top: standard tracking, where fast motion and tight boxes cause drift and ID switches. Bottom: our method modifies $t_2$, where $t_2' \leftarrow t_2$ after `DampenVelocity` and $t_2'' \leftarrow t_2'$ after `EnlargeBox`, preserving consistent identities during occlusion.

**Tracklet Initialization and Deletion.** To maintain a clean and reliable set of tracklets, we apply explicit initialization and deletion policies. Remaining high-confidence detections are considered for initialization only if they have low overlap with existing tracklets, i.e., $\max_{t \in \mathcal{T}} \text{IoU}(d, t) < K_{\text{init}}$, ensuring that redundant or duplicate tracks are avoided. On the other hand, unmatched tracklets in $\mathcal{T}_{\text{remain}}$ are deleted unless they are marked as occluded. For occluded tracklets, we allow temporary persistence but delete them if their occlusion age exceeds a threshold $T_{\text{occ}}$. This strategy ensures long-term robustness while avoiding stale or spurious tracks.

**Post Processing** While our method is designed for fully online inference with minimal reliance on post-processing, we also incorporate two complementary post-processing techniques to showcase their potential benefits. First, global linking is used to associate fragmented tracklets by leveraging spatiotemporal consistency and appearance features extracted via GIAOTracker's (Du et al., 2021) ResNet50-TP encoder, with tracklet-level matching based on cosine similarity. Second, Gaussian Smoothing Process (GSP) (Schulz et al., 2018) is applied to refine tracklet trajectories by modeling nonlinear motion over time. Unlike linear interpolation, GSP incorporates both past and future observations, offering more robust handling of missing detections and smoother trajectory corrections.

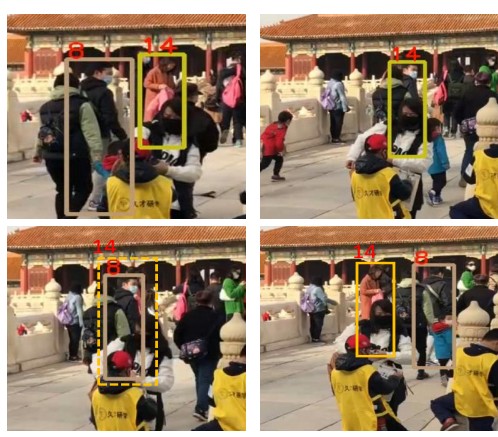

Figure 6: Effectiveness of bounding box enlargement in occlusion handling. Top row: baseline tracker fails to maintain ID during occlusion. Bottom row: enlarged bounding box during occlusion enables successful ID recovery after reappearance.

**Benchmark.** To comprehensively evaluate multi-object tracking in complex traffic scenes, we introduce the FastTrack benchmark—a diverse and challenging dataset that surpasses existing benchmarks such as UrbanTracker and CityFlow in several key dimensions. FastTrack contains 800K annotated detections across 12 videos, each densely populated with an average of 43.5 objects per frame—more than 5× that of UrbanTracker and over 5× CityFlow—making it particularly suitable for evaluating trackers under extreme crowding and interaction.

| Attribute | UrbanTracker | CityFlow | FastTracker |
|---|---|---|---|
| Year | 2014 | 2022 | 2025 |
| Detections | 12.5K | 890K | 800k |
| #Videos | 5 | 40 | 12 |
| Obj./Frame | 5.4 | 8.2 | 43.5 |
| #Classes | 3 | 1 | 9 |
| #Scenarios | 1 | 4 | 12 |

Table 1: Comparison of traffic multi-object tracking datasets. 'Obj./Frame' denotes the average number of objects per frame.

The dataset spans 9 traffic-related classes, expanding the label diversity beyond those in prior datasets. Furthermore, FastTrack encompasses 12 distinct traffic scenarios, including multilane intersections, crosswalks, tunnels, and merging roads, under varied lighting conditions such as daylight, night scenes, and strong shadow transitions. These factors introduce frequent and severe occlusions, challenging trackers to maintain identity continuity even during long-term disappearances. Compared to existing datasets, which often feature limited scene types and low object density, FastTrack provides a much more realistic and exhaustive benchmark for modern tracking algorithms, especially those designed for deployment in urban traffic environments. Benchmark statistics and visualizations are in Table 1 and Figure 7 respectively. Annotation and split setting are provided in Appendix D.

## 5 EXPERIMENT

To evaluate the effectiveness and robustness of our proposed tracking framework, we conduct extensive experiments across both standard benchmarks and internal studies. First, we perform a comprehensive ablation study to investigate the contribution of each individual component in our pipeline. Then, we present quantitative results on public multi-object tracking benchmarks including MOT16, MOT17, MOT20, DanceTrack, BDD100K and our newly added benchmark dataset, highlighting the competitiveness of our method in both crowded and occluded scenarios. For completeness, the intuitive Python code and full algorithm description are in Appendix C. Additional details on the datasets, evaluation metrics, hyperparameter choices, and experimental setup are included in Appendix D. Complexity analysis is in Appendix E.

Figure 7: Sample frames from the Fast-Track benchmark. The dataset captures complex interactions, high object density, and frequent occlusions.

### 5.1 ABLATION STUDIES

To examine the impact of each core component in our framework, we perform a series of ablation studies on validation sets. Specifically, we examine four major aspects: (1) the effect of our deletion and initialization policies on maintaining track consistency, (2) the contribution of our occlusion-aware mechanism in preserving identities through temporary visual loss, (3) the influence of incorporating user-defined ROI and directional constraints (cone-based filtering), (4) the role of post-processing techniques such as global linking and trajectory smoothing, and (5) check the performance of FastTracker on lighter detectors.

**Deletion and Initialization Policies:** As shown in Table 2, enabling either deletion or initialization independently brings modest gains, while their combination consistently yields the best performance. Specifically, on MOT17, jointly applying both strategies improves MOTA from 79.4 to 79.9 and HOTA from 63.5 to 64.0, resulting in absolute gains of +0.5 MOTA and +0.5 HOTA. Similarly, on MOT20, we observe a +0.7 MOTA and +0.5 HOTA improvement. For the more challenging FastTrack dataset, the joint configuration boosts MOTA from 60.1 to 60.9 and HOTA from 57.2 to 58.0. These results confirm the effectiveness of our proposed deletion and initialization policies. Further ablations on the impact of $K_{init}$ are provided in Table 14 of the Appendix.

| Del | Init | MOT17-val | | MOT20-val | | FastTrack | |
|---|---|---|---|---|---|---|---|
| | | MOTA ↑ | HOTA ↑ | MOTA ↑ | HOTA ↑ | MOTA ↑ | HOTA ↑ |
| ✓ | | 79.4 | 63.5 | 74.5 | 62.8 | 60.1 | 57.2 |
| | ✓ | 79.6 | 63.7 | 74.6 | 63.0 | 60.4 | 57.6 |
| ✓ | ✓ | **79.9** | **64.0** | **75.2** | **63.3** | **60.9** | **58.0** |

Table 2: Ablation study on deletion and initialization policies across three datasets.

**Occlusion Handling:** To isolate the effect of occlusion handling, we disable our proposed deletion and initialization policies and revert to the conventional strategy—removing tracklets immediately when not visible and initializing new ones from all remained high-confidence

| Occ | MOT17-val | | MOT20-val | | FastTrack | |
|---|---|---|---|---|---|---|
| | MOTA ↑ | HOTA ↑ | MOTA ↑ | HOTA ↑ | MOTA ↑ | HOTA ↑ |
| | 79.0 | 63.1 | 74.1 | 62.1 | 59.2 | 56.8 |
| ✓ | **80.4** | **65.2** | **76.7** | **64.4** | **63.4** | **60.7** |

Table 3: Ablation study on occlusion handling.

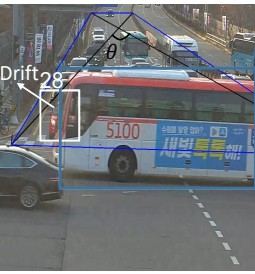 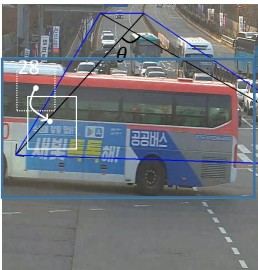 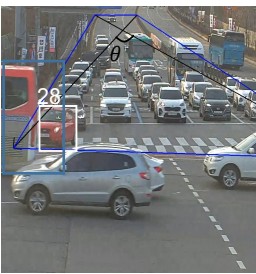

Figure 8: Illustration of ROI and direction constraints. Frame 2: Target ID 28 is occluded by ID 51, causing its trajectory to drift outside the valid region. Frame 3: the direction limitation and ROI constraint modules (`ProjectToCone`, `ClampToROI`) correct the trajectory back into the allowable region. Frame 4: ID 28 is recovered. Failure cases of baselines are in Appendix F.5

detections. As shown in Table 3, incorporating our occlusion-aware mechanism yields a notable performance gain across all datasets. Specifically, on MOT17, HOTA improves by 2.1 points (from 63.1 to 65.2) and MOTA by 1.4 points (from 79.0 to 80.4). On the more crowded MOT20 dataset, our approach increases HOTA by 2.3 points and MOTA by 2.6 points. The largest gains are observed on FastTrack, where HOTA improves by 3.9 points and MOTA by 4.2 points. For completeness, we additionally provide ablation studies on $\text{IoU}_{\min}$, $T_{\text{occ}}$, $\beta_{\text{enlarge}}$, $\gamma_{\text{velo}}$, and $(\delta_p, \delta_v)$ with results summarized in Tables 15, 16, 17, 18, and 19 in the Appendix.

**ROI and Direction.** To further refine the association process in structured scenes, we examine the effect of incorporating Region-of-Interest (ROI) filtering and direction (Dir) constraints on the FastTrack dataset, as shown in Table 4. These evaluations are conducted on top of other modules, including the proposed occlusion handling and deletion/initialization policies. Applying the ROI constraint alone achieves 63.7 MOTA / 61.0 HOTA, while the direction constraint improves the scores to 64.2 MOTA / 61.3 HOTA. When both constraints are combined, the method reaches the best result of 64.4 MOTA / 61.5 HOTA. A qualitative example of these effects is illustrated in Figure 8.

| | | FastTrack | |
|---|---|---|---|
| ROI | Dir | MOTA ↑ | HOTA ↑ |
| ✓ | | 63.7 | 61 |
| | ✓ | 64.2 | 61.3 |
| ✓ | ✓ | **64.4** | **61.5** |

Table 4: Ablation study on ROI and direction constraint (Dir) using the FastTrack dataset.

**Post Processing:** To evaluate the contribution of the post-processing stage, we analyze the impact of incorporating the GSP, and global linking (G-Link) on the final tracking performance across three datasets. Importantly, all results reported here are based on the full online system, which includes our proposed deletion/initialization policies and occlusion handling mechanisms. The post-processing modules are applied offline after generating the initial tracks. As shown in Table 5, enabling either GSP or G-Link individually brings consistent improvements over the base output, with G-Link slightly outperforming GSP in terms of HOTA across all datasets.

**Tracking performance of light modified YOLOX.** FastTracker demonstrates robust performance even when paired with lightweight YOLOX detectors as shown in Table 6. As the model size decreases from YOLOX-L (61M) to YOLOX-Nano (1M), the tracking accuracy degrades gracefully, maintaining strong metrics across the board. Notably, YOLOX-M and YOLOX-S achieve MOTA scores of 78.1 and 74.6 respectively, outperforming several base-

| | | MOT17-val | | MOT20-val | | FastTrack | |
|---|---|---|---|---|---|---|---|
| GSP | G-Link | MOTA ↑ | HOTA ↑ | MOTA ↑ | HOTA ↑ | MOTA ↑ | HOTA ↑ |
| ✓ | | 82.0 | 66.5 | 80.1 | 65.9 | 64.7 | 61.9 |
| | ✓ | 82.0 | 66.6 | 80.1 | 66 | 64.8 | 62 |
| ✓ | ✓ | **82.2** | **66.6** | **80.3** | **66.3** | **65** | **62.2** |

Table 5: Ablation study on global spatial prior (GSP) and global linking (G-Link) across three datasets.

lines that rely on larger detectors and heavier re-identification modules. Even with YOLOX-Nano, FastTracker achieves a reasonable MOTA of 68.3 and IDF1 of 71.2, enabling real-time deployment on edge devices with limited resources. These results highlight the efficiency and scalability of FastTracker across a range of detector capacities.

## 5.2 BENCHMARK EVALUATION

In the final part of our experiments, we evaluate the effectiveness of our proposed tracker against state-of-the-art methods on MOT16, MOT17, and MOT20 testsets and also Dance-Track, BDD100K and FastTrack benchmarks. To demonstrate our online tracking performance, we submit our tracking outputs (without any post-processing such as GSP or G-Link) to the respective evaluation servers. This setup ensures that all reported results reflect the performance of our online system only, highlighting the strength of our tracking framework under standard benchmark protocols and constraints.

| Detector | Params | MOTA ↑ | HOTA ↑ | IDF1 ↑ |
|---|---|---|---|---|
| YOLOX-L | 61M | 79.3 | 65.1 | 79.8 |
| YOLOX-M | 28M | 78.1 | 64.0 | 77.6 |
| YOLOX-S | 10.5M | 74.6 | 61.7 | 74.7 |
| YOLOX-Tiny | 5.8M | 73.9 | 60.2 | 74.1 |
| YOLOX-Nano | 1M | 68.3 | 56.8 | 71.2 |

Table 6: Performance of FastTracker using lightweight YOLOX models on the MOT17 validation set.

**MOT16 and MOT17:** We evaluate FastTracker on the official MOT16 benchmark testset, where it achieves strong performance with a MOTA of 79.1 and HOTA of 66.0. Full comparisons with baseline methods are provided in Table 20 in the Appendix.

Table 7: MOT17: Performance comparison with SOTA. Best: **Bold** blue, Second best: Red.

| Method | MOTA↑ | HOTA↑ | IDF1↑ | FP↓ | FN↓ | IDs↓ |
|---|---|---|---|---|---|---|
| CTracker (Peng et al., 2020) | 66.6 | 49.0 | 57.4 | 22284 | 160491 | 5529 |
| CenterTrack (Zhou et al., 2020) | 67.8 | 60.3 | 64.7 | 18489 | 160332 | 3039 |
| QDTrack (Pang et al., 2021) | 68.7 | 63.5 | 66.3 | 26598 | 146643 | 3378 |
| TraDes (Wu et al., 2021) | 69.1 | 52.7 | 63.9 | 20892 | 150060 | 3555 |
| SOTMOT (Zheng et al., 2021) | 71 | 64.1 | 71.9 | 39537 | 118983 | 5184 |
| GSDT (Wang et al., 2021b) | 73.2 | 55.2 | 66.5 | 26397 | 120666 | 3891 |
| RelationTrack (Yu et al., 2022) | 73.8 | 59.9 | 74.7 | 27999 | 118623 | 1374 |
| TransTrack (Sun et al., 2020b) | 74.5 | 54.1 | 63.9 | 28323 | 112137 | 3663 |
| OMC–F (Liang et al., 2022a) | 74.7 | 56.8 | 73.8 | 30162 | 108556 | - |
| CSTrack (Liang et al., 2022b) | 74.9 | 59.3 | 72.3 | 23847 | 114303 | 3567 |
| OMC (Liang et al., 2022a) | 76 | 57.1 | 73.8 | 28894 | 101022 | - |
| SGT (Hyun et al.) | 76.3 | 57.3 | 72.8 | 25974 | 102885 | 4101 |
| CorrTracker (Wang et al., 2021a) | 76.4 | 58.4 | 73.6 | 29808 | 99510 | 3369 |
| FairMOT (Zhang et al., 2021) | 73.7 | 59.3 | 72.3 | 27507 | 117477 | 3303 |
| DeepSORT (Wojke et al., 2017) | 78 | 61.2 | 74.5 | 29852 | 94716 | 1821 |
| ByteTrack (Zhang et al., 2022) | 78.9 | 62.8 | 77.2 | 25491 | 83721 | 2196 |
| StrongSORT (Du et al., 2023) | 78.3 | 63.5 | 78.5 | 27876 | 86205 | 1446 |
| FeatureSORT (Hashempoor et al., 2024) | 79.6 | 63 | 77.2 | 29588 | 83132 | 2269 |
| FLWM (Liu et al., 2024) | 80.5 | 64.9 | 79.9 | 27245 | 81653 | 1370 |
| SparseTrack (Liu et al., 2025) | 81 | 65.1 | 80.1 | 23904 | 81927 | 1170 |
| PD-SORT (Wang et al., 2025) | 79.3 | 63.9 | 79.2 | 17028 | 101130 | 1062 |
| C-TWiX (Miah et al., 2025) | 78.1 | 63.1 | 76.3 | 20964 | 96642 | 5820 |
| TOPICTrack (Cao et al., 2025) | 78.8 | 63.9 | 78.6 | 17010 | 101130 | 1515 |
| EscapeTrack (Yi et al., 2025) | 80.8 | 66.2 | 81.9 | 34908 | 75252 | 1061 |
| FastTracker | **81.8** | **66.4** | **82.0** | 26850 | **75162** | **885** |

Table 8: MOT20: Performance comparison with SOTA. Best: **Bold** blue, Second best: Red.

| Method | MOTA↑ | HOTA↑ | IDF1↑ | FP↓ | FN↓ | IDs↓ |
|---|---|---|---|---|---|---|
| SORT (Bewley et al., 2016) | 42.7 | 36.1 | 45.1 | 28398 | 287582 | 4852 |
| Tracktor++ (Bergmann et al., 2019) | 52.6 | 42.1 | 52.7 | 35536 | 236680 | 1648 |
| CSTrack (Liang et al., 2022b) | 66.6 | 54 | 68.6 | 25404 | 144358 | 3196 |
| CrowdTrack (Stadler & Beyerer, 2021) | 70.7 | 55 | 68.2 | 21928 | 126533 | 3198 |
| RelationTrack (Yu et al., 2022) | 67.2 | 56.5 | 70.5 | 61134 | 104597 | 4243 |
| DeepSORT (Wojke et al., 2017) | 71.8 | 57.1 | 69.6 | 37858 | 101581 | 3754 |
| TransTrack (Sun et al., 2020b) | 64.5 | 48.9 | 59.2 | 28566 | 151377 | 3565 |
| CorrTracker (Wang et al., 2021a) | 65.2 | 57.1 | 69.1 | 79429 | 95855 | 5193 |
| GSDT (Wang et al., 2021b) | 67.1 | 53.6 | 67.5 | 31913 | 135409 | 3131 |
| SOTMOT (Zheng et al., 2021) | 68.6 | 55.7 | 71.4 | 57064 | 101154 | 4209 |
| OMC (Liang et al., 2022a) | 73.1 | 60.5 | 74.4 | 16159 | 108654 | 779 |
| FairMOT (Zhang et al., 2021) | 61.8 | 54.6 | 67.3 | 103404 | 88901 | 5243 |
| SGT (Hyun et al.) | 64.5 | 56.9 | 62.7 | 67352 | 111201 | 4909 |
| ByteTrack (Zhang et al., 2022) | 75.7 | 60.9 | 74.9 | 26249 | 87594 | 1223 |
| StrongSORT (Du et al., 2023) | 72.2 | 61.5 | 75.9 | 16632 | 117920 | 770 |
| FeatureSORT (Hashempoor et al., 2024) | 76.6 | 61.3 | 75.1 | 25083 | 95027 | 1081 |
| FLWM (Liu et al., 2024) | 77.7 | 62 | 75 | 25019 | 88959 | 1530 |
| SparseTrack (Liu et al., 2025) | **78.2** | 63.4 | 77.3 | 25108 | 86720 | 1116 |
| PD-SORT (Wang et al., 2025) | 75.4 | 62.6 | 76.7 | 23974 | 93662 | 908 |
| C-TWiX (Miah et al., 2025) | 75 | 62.4 | 74.1 | 25933 | 115240 | 2048 |
| TOPICTrack (Cao et al., 2025) | 72.4 | 62.6 | 77.6 | 10986 | 131088 | 869 |
| EscapeTrack (Yi et al., 2025) | 77.4 | 64.8 | 80.5 | 30666 | 85188 | 1061 |
| FastTracker | 77.9 | 65.7 | 81.0 | 24590 | 89243 | **684** |

Table 7 shows the performance of FastTracker on the MOT17 benchmark testset. Our method achieves a new state-of-the-art with MOTA of 81.8 and HOTA of 66.4, outperforming previous trackers. Compared to FeatureSORT (MOTA 79.6, HOTA 63.0), we achieve improvements of +2.2 MOTA and +3.4 HOTA. Against StrongSORT (MOTA 78.3, HOTA 63.5), the gains are +3.5 MOTA and +2.9 HOTA. Even the widely adopted ByteTrack shows lower performance (MOTA 78.9, HOTA 62.8), with a margin of +2.9 MOTA and +3.6 HOTA. Additionally, FastTracker achieves the lowest number of ID switches (885), confirming both strong identity preservation and detection quality.

**MOT20:** Table 8 presents the results on the challenging MOT20 benchmark testset. FastTracker is among the top performances across most key metrics with MOTA 77.9, HOTA 65.7, and IDF1 81.0. This represents a significant improvement over prior methods. Compared to FeatureSORT (MOTA 76.6, HOTA 61.3, IDF1 75.1), we see gains of +1.3 MOTA, +4.4 HOTA, and +5.9 IDF1. Against ByteTrack (MOTA 75.7, HOTA 60.9, IDF1 74.9), our method improves by +2.2 MOTA, +4.8 HOTA, and +6.1 IDF1. Notably, FastTracker also achieves the lowest ID switches (684) among all trackers, indicating robust identity preservation even in extremely crowded scenes. These results confirm that FastTracker delivers state-of-the-art tracking performance in high-density environments relying on its occlusion handling capabilities.

| Method | MOTA↑ | HOTA↑ | IDF1↑ |
|---|---|---|---|
| TraDes (Wu et al., 2021) | 86.2 | 43.3 | 41.2 |
| SGT (Hyun et al.) | 85.1 | 43.7 | 47.2 |
| DeepSORT (Wojke et al., 2017) | 88.2 | 46.4 | 51.7 |
| FairMOT (Zhang et al., 2021) | 82.2 | 39.7 | 40.8 |
| ByteTrack (Zhang et al., 2022) | 89.6 | 47.7 | 53.9 |
| StrongSORT (Du et al., 2023) | 91.1 | 55.6 | 55.2 |
| FLWM (Liu et al., 2024) | 90 | 62.2 | 61.4 |
| UCMCTrack (Yi et al., 2024) | 88.9 | 65 | 63.6 |
| SparseTrack (Liu et al., 2025) | 91.3 | 55.5 | 58.3 |
| PD-SORT (Wang et al., 2025) | 87.4 | 55.5 | 55.4 |
| C-TWiX (Miah et al., 2025) | 91.4 | 62.2 | 63.5 |
| TOPICTrack (Cao et al., 2025) | 89.3 | 55.9 | 54.5 |
| EscapeTrack (Yi et al., 2025) | 89.5 | 62 | 66.4 |
| FastTracker | **93.4** | **65.9** | 67.2 |

Table 9: DanceTrack: Performance comparison with SOTA. Best: **Bold** blue, Second best: Red.

**DanceTrack and BDD100K.** We include comparison with the DanceTrack to demonstrate that FastTracker generalizes well beyond the MOT benchmarks. As shown in Table 9, FastTracker achieves the highest scores with 93.4 MOTA and 65.9 HOTA, outperforming prior methods. All

ablation studies for DanceTrack are conducted and provided in Appendix. Fasttracker achieves 43.8 mMOTA and 56.2 mIDF1 on BDD100K, with full comparison tables provided in the Appendix.

**FastTrack Benchmark:** Table 10 presents the results on the FastTrack Benchmark, a challenging internal benchmark designed to assess robustness in dense tracking scenarios. FastTracker achieves the best MOTA (64.4), IDF1 (79.2), and HOTA (61.5), surpassing FeatureSORT and ByteTrack in both detection and association accuracy. It also records the lowest number of identity switches (251), demonstrating strong consistency across frames.

Compared to StrongSORT and FeatureSORT, FastTracker improves HOTA by +3.5 and +1.8 points, respectively, while significantly reducing ID switches. These results highlight FastTracker's strength in maintaining accurate identities under crowded and occluded conditions. Figure 9 illustrates the relationship between occlusion levels in 4 FastTrack benchmark videos and identity switches. The percentages shown in the bars denote the

| Method | MOTA↑ | HOTA↑ | IDF1↑ | FP↓ | FN↓ | IDs↓ |
|---|---|---|---|---|---|---|
| ByteTrack (Zhang et al., 2022) | 56.8 | 60.0 | 77.5 | 3656 | 68552 | 533 |
| StrongSORT (Du et al., 2023) | 61.1 | 58.0 | 77.0 | 33989 | **64882** | 422 |
| FeatureSORT (Hashempoor et al., 2024) | 61.4 | 59.7 | 77.4 | **26993** | 74692 | 390 |
| SparseTrack (Liu et al., 2025) | 61.9 | 59.9 | 77.4 | 30652 | 65874 | 397 |
| PD-SORT (Wang et al., 2025) | 61.6 | 60.3 | 77 | 28764 | 70998 | 338 |
| FastTracker | **64.4** | **61.5** | **79.2** | 29730 | 68541 | **251** |

Table 10: FastTrack benchmark comparison with SOTA. Best in **bold**.

relative reduction in ID switches of FastTracker w.r.t. the baseline ByteTrack, indicating the outperformance of FastTracker. Notably, higher reductions are observed in videos with more occluded objects, highlighting the robustness of our method under challenging conditions.

**Qualitative Results.** We further provide qualitative visualizations in the Appendix F.5, where multiple examples illustrate the effectiveness of our proposed modules—such as ROI and direction constraints, velocity damping, and bounding-box enlargement—in recovering occluded targets and maintaining consistent identities.

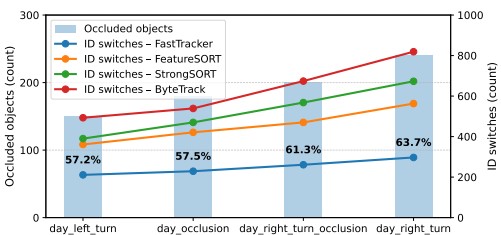

Figure 9: Reduction in ID switches vs. baselines. Bars: occluded objects (left axis). Lines: ID switches per method (right axis).

## 6 DISCUSSION

FastTracker operates in real time across all evaluated scenarios, sustaining frame-rate processing on the incoming streams even under severe occlusions and dense object/frame conditions. This efficiency enables deployment on a wide range of edge devices without sacrificing accuracy. While FastTracker demonstrates strong performance across public and internal benchmarks, some limitations remain. Currently, the system relies on manually defined ROI regions and cone direction constraints, which must be specified using exactly four edges. This configuration may limit deployment in complex or dynamic scenes where such annotations are impractical or insufficient. As a promising direction for future work, recent advances in semantic segmentation and scene understanding can be leveraged to enable automatic detection of road boundaries, crosswalks, and other contextual cues, removing the need for manual setup. Moreover, extending the system to support arbitrary polygonal ROIs or more flexible directional constraints would allow broader applicability in real-world environments such as intersections, roundabouts, and multi-lane roads.

## 7 CONCLUSION

We introduce FastTracker, a very fast and lightweight multi-object tracker that operates without any CNN-based re-identification network. It effectively handles occlusions and leverages environment-aware cues such as spatial constraints to improve tracking accuracy. While designed for online deployment, it also supports optional post-processing for further refinement. Despite its simplicity, FastTracker achieves consistently high scores—for example, MOTA above 79 on MOT16/17 and over 93 on DanceTrack, with strong HOTA improvements across all benchmarks—surpassing several state-of-the-art methods. These results highlight its robustness under both crowded MOT scenarios and diverse non-MOT benchmarks, making FastTracker particularly suitable for real-world deployment on resource-constrained devices.

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

APPENDIX

CONTENTS OF APPENDIX

## A  RELATED WORK

In the tracking-by-detection paradigm, object detections—typically generated by deep convolutional networks (Sun et al., 2021)—are first obtained and then associated across frames to form trajectories. Many frameworks compute similarity between detections and existing tracklets using a combination of geometric and appearance-based cues. While appearance features have been widely adopted, especially through deep learning methods such as DeepSORT (Wojke et al., 2017) and FeatureSORT or their extensions employing state space models (SSM) (Hashempoorikderi & Choi, 2024; Hashempoor & Choi, 2025) such as (Wang et al., 2024a), they require additional re-identification (Re-ID) networks which significantly increase computational load and memory usage. This added complexity often limits their suitability for real-time or resource-constrained applications.

In contrast, several lightweight tracking methods avoid deep appearance models and instead rely on hand-crafted or geometry-based features, especially in multi-class or traffic-heavy environments where speed and scalability are essential. One of the most notable examples is SORT (Bewley et al., 2016), which uses only motion-based association via the Hungarian algorithm, offering impressive speed and simplicity. Recent works have shown renewed interest in such efficient designs. For instance, OC-SORT (Cao et al., 2023) enhances traditional motion-only tracking with improved observation consistency modeling, while BoT-SORT (Aharon et al., 2022) introduces a modular framework that decouples tracking logic from heavy feature extraction. These approaches demonstrate that robust tracking can be achieved without deep appearance embeddings, particularly when runtime efficiency is a priority.

For occlusion handling, efficient methods like ByteTrack address partial occlusions by associating low-confidence detections, but performance degrades in crowded scenes where occluded detections become ambiguous. Recent approaches integrate appearance-based re-identification features using CNNs (Zhang et al., 2023) for better identity recovery, but these incur high computational overhead, making them unsuitable for real-time or resource-constrained applications. Lightweight alternatives like PD-SORT (Wang et al., 2025) and SparseTrack (Liu et al., 2025) use geometry-driven strategies with pseudo-depth cues from 2D bounding boxes. However, these rely on camera viewpoint assumptions and simple depth heuristics that can fail under non-ideal perspectives or rapid scene changes. Additionally, depth-based association struggles with similar depths or long-term occlusion, leading to identity drift. We propose an occlusion-handling mechanism that avoids viewpoint assumptions and depth-based matching, addressing the limitations of perspective distortion and ambiguous proximity in crowded scenes.

Prior works have explored the use of environmental context to support object tracking, though typically in limited or indirect ways. For example, (Sani & Anand, 2024) and MENet (Huang et al., 2023) employ predefined Regions of Interest (ROIs) to restrict tracking to drivable areas,

effectively discarding detections outside these zones. However, these methods do not correct or adapt tracklets that drift beyond the ROIs due to noise or occlusion. In the domain of multi-camera tracking, many approaches leverage contextual zones—such as entry and exit regions—to guide inter-camera association (e.g. Chai et al. (Chai et al., 2024)), but such strategies generally do not make use of fine-grained, single-camera environmental layout to refine trajectories. In contrast, our method incorporates high-level scene structures such as road directions, two-way traffic, and pedestrian crosswalks to adapt tracklets that violate feasible motion patterns. This environment-aware correction is geometry-driven, lightweight, and operates without relying on CNN-based modules, enabling improved tracking consistency and reduced ID switches in urban scenes with complex layout constraints.

Several benchmarks have been introduced for vehicle and multi-class object tracking, but each has limitations in real-world urban surveillance contexts. The Waymo Open Dataset(Sun et al., 2020a) provides large-scale LiDAR and camera data for autonomous driving, with high-quality labels, but it focuses primarily on front-facing vehicle-mounted views and highway scenarios, which differ from urban CCTV perspectives. KITTI Tracking(Geiger et al., 2012) offers annotations for cars, pedestrians, and cyclists, but lacks diversity in scenes, especially in crowded urban intersections. LMOT(Wang et al., 2024b) introduces a nighttime benchmark under low-light conditions but lacks multi-class vehicle tracking and diverse lighting scenarios. VETRA(Hellekes et al., 2024) provides aerial vehicle tracking data with scale variation but is limited to single-class tracking and overhead views, which are less relevant to ground-based systems. In contrast, our benchmark—while moderate in size—focuses on multi-class tracking from urban CCTV viewpoints, which are underrepresented in existing datasets. It includes diverse object categories, varied lighting, and challenging occlusion scenarios like dense intersections and bidirectional traffic, making it a more practical resource for developing and evaluating trackers in city-scale applications.

## B   Kalman Motion Model

**State Representation.** We use a standard 8-dimensional Kalman filter commonly adopted in MOT frameworks for modeling 2D bounding-box motion in image space. The state vector at time $t$ is

$$s_t = (x, y, a, h, v_x, v_y, v_a, v_h)^\top,$$

where $(x, y)$ is the center of the bounding box, $a$ is the aspect ratio, $h$ is the height, and $(v_x, v_y, v_a, v_h)$ are their corresponding velocities. We assume a constant-velocity motion model, and the observable part of the state is the bounding-box geometry $(x, y, a, h)$, which forms a linear observation of the state.

**Prediction Step.** The Kalman filter proceeds in two stages. In the prediction stage, the state is propagated using a class-specific linear transition matrix $\mathbf{G}_{\mathrm{cls}}$, whose time-step parameter $dt$ depends on the object category (see Table 11). The prediction mean and covariance are:

$$\mu_{t|t-1} = \mathbf{G}_{\mathrm{cls}}\, \mu_{t-1|t-1}, \tag{4}$$

$$\mathbf{P}_{t|t-1} = \mathbf{G}_{\mathrm{cls}}\, \mathbf{P}_{t-1|t-1}\, \mathbf{G}_{\mathrm{cls}}^\top + \mathbf{Q}, \tag{5}$$

where $\mu_{t-1|t-1}$ and $\mathbf{P}_{t-1|t-1}$ are the previous filtered state mean and covariance, and $\mathbf{Q}$ is the process-noise covariance. Making $\mathbf{G}$ class-dependent allows faster-moving classes (e.g., cars, motorcycles) and slower classes (e.g., pedestrians) to follow more realistic motion ranges.

**Update Step.** Given a measurement $\mathbf{z}_t = (x, y, a, h)^\top$, the Kalman gain is computed as:

$$\mathbf{K}_t = \mathbf{P}_{t|t-1}\, \mathbf{H}^\top \left(\mathbf{H}\, \mathbf{P}_{t|t-1}\, \mathbf{H}^\top + \mathbf{R}\right)^{-1}, \tag{6}$$

where $\mathbf{H}$ selects the observable components of the state and $\mathbf{R}$ is the measurement-noise covariance.

The updated (filtered) state becomes:

$$\mu_{t|t} = \mu_{t|t-1} + \mathbf{K}_t \left(\mathbf{z}_t - \mathbf{H}\, \mu_{t|t-1}\right), \tag{7}$$

$$\mathbf{P}_{t|t} = \left(\mathbf{I} - \mathbf{K}_t \mathbf{H}\right) \mathbf{P}_{t|t-1}. \tag{8}$$

**NSA Kalman Filter (Noise-Scale Adaptation).** While the standard Kalman filter provides a stable motion model, it assumes a fixed measurement-noise covariance $\mathbf{R}$ and therefore reacts poorly to imperfect or low-confidence detections. To make the update step more robust, we adopt the Noise-Scale Adaptation (NSA) Kalman filter (Du et al., 2021), which adjusts the observation-noise covariance based on the confidence of each detection.

Given the detection confidence $Conf_t \in [0, 1]$ at time $t$, the adaptive measurement-noise covariance is defined as

$$\hat{\mathbf{R}}_t = \left(1 - Conf_t\right) \mathbf{R}, \tag{9}$$

where $\mathbf{R}$ is the base measurement-noise covariance (typically constant in the standard Kalman model).

This formulation means that high-confidence detections (large $Conf_t$) are treated as more reliable and receive a smaller $\hat{\mathbf{R}}_t$, giving them greater influence during the update step. Conversely, low-confidence detections are treated as noisier, reducing their impact on the filtered state.

All Kalman filter equations described earlier remain unchanged; the only modification is replacing the fixed $\mathbf{R}$ in the update equations with the confidence-adaptive $\hat{\mathbf{R}}_t$.

**Category-Aware Kalman Motion Model.** Following the description in the main paper, our motion model is class-aware: the Kalman prediction step adapts its transition dynamics according to the expected speed range of each object category. The only parameter that changes across categories is the time-step factor $dt$ in the state-transition matrix $\mathbf{G}_{cls} = \mathbf{G}(dt)$:

$$\mathbf{G}(dt) = \begin{bmatrix} 1 & 0 & 0 & 0 & dt & 0 & 0 & 0 \\ 0 & 1 & 0 & 0 & 0 & dt & 0 & 0 \\ 0 & 0 & 1 & 0 & 0 & 0 & dt & 0 \\ 0 & 0 & 0 & 1 & 0 & 0 & 0 & dt \\ 0 & 0 & 0 & 0 & 1 & 0 & 0 & 0 \\ 0 & 0 & 0 & 0 & 0 & 1 & 0 & 0 \\ 0 & 0 & 0 & 0 & 0 & 0 & 1 & 0 \\ 0 & 0 & 0 & 0 & 0 & 0 & 0 & 1 \end{bmatrix}.$$

A larger $dt$ allows a tracklet to move faster between frames, while a smaller $dt$ enforces slower and smoother motion. This provides realistic category-aware trajectory prediction as detailed in table 11, especially during short occlusions or detector dropouts.

Table 11: Category-specific time-step parameter $dt$ for the class-aware Kalman predictor.

| Category | $dt$ |
|---|---|
| person | 0.20 |
| bike | 0.35 |
| pm (scooter) | 0.40 |
| motorbike | 0.55 |
| car | 1.00 |
| truck_small | 0.85 |
| truck_big | 0.80 |
| bus_small | 0.75 |
| bus_big | 0.70 |

## C  ALGORITHM AND PYTHON INTUITIVE CODE

### C.1  PYTHON INTUITIVE CODE

In the following, we provide an intuitive Python-style pseudocode that complies with the logic described in the main body of the paper. Most of the core ideas can be expressed in just a few lines, highlighting the simplicity and modularity of our approach.

```python
# FastTracker (intuitive pseudocode in Python)

def ProjectToCone(state, cone):
    """
    Adjust velocity direction to lie within cone
    """
    # Eq. 1
    state.vx, state.vy = adjust_to_cone(state.vx, state.vy, cone)
    return state

def DampenVelocity(track, gamma=0.5, delta_v=1):
    """
    Apply velocity damping during occlusion.
    """
    # Eq. 2
    track.vx = gamma * track.history[-delta_v].vx
    track.vy = gamma * track.history[-delta_v].vy
    return track

def EnlargeBox(track, scale=1.2):
    """
    Enlarge bounding box to account for uncertainty.
    """
    # Eq. 3
    track.w *= scale
    track.h *= scale
    return track

class FastTracker:
    def __init__(self, env_map, cp_min, t_occ, k_init):
        self.tracks = []
        self.env_map = env_map
        self.cp_min = cp_min
        self.t_occ = t_occ
        self.k_init = k_init

    def update(self, detections):
        # Split detections by confidence
        D_high = [d for d in detections if d.conf > h_thresh]
        D_low  = [d for d in detections if l_thresh < d.conf <= h_thresh]

        # Prediction with environment constraints
        for t in self.tracks:
            state = kalman_predict(t)
            region = lookup_region(self.env_map, state)
            if region:
                state = ProjectToCone(state, region.cone)
                state = clamp_to_roi(state, region)
            t.state = state

        # Stage 1 & 2 association (simplified)
        T1, D1 =associate(self.tracks, D_high)
        T2, D2 =associate([t for t in self.tracks if t not in T1], D_low)
        self.tracks = T1 + T2

        # Occlusion handling
        T_remain = [t for t in self.tracks if t not in (T1+T2)]
        T_occ = [t for t in T_remain if max_IoU(t, self.tracks) >= self.
    cp_min]
        for t in T_occ:
            t.occluded = True
            DampenVelocity(t)
```

---

**Algorithm 1** FastTracker

---

**Require:** Video $V$; detector Det; thresholds $\tau_{\text{high}} > \tau_{\text{low}}$; environment map $\mathcal{M}$; occlusion params
   $\text{IoU}_{\text{min}}, T_{\text{occ}}$; init/deletion params $K_{\text{init}}$
**Ensure:** Tracks $\mathcal{T}$
 1: $\mathcal{T} \leftarrow \emptyset$
 2: **for** each frame $f_k$ in $V$ **do**
 3:    $\mathcal{D}_{\text{high}}, \mathcal{D}_{\text{low}} \leftarrow \text{Det}(f_k)$
      // Prediction with map-based constraints
 4:    **for** each $t \in \mathcal{T}$ **do**
 5:       $\hat{s}_t \leftarrow \text{KalmanPredict}(t)$
 6:       $R \leftarrow \text{RegionLookup}(\mathcal{M}, \hat{s}_t)$
 7:       **if** $R \neq \varnothing$ **then**
            // Direction constraint
 8:          $\hat{s}_t \leftarrow \text{ProjectToCone}(\hat{s}_t, \text{cone}(R))$          ( equation 1)
            // Stay inside ROI
 9:          $\hat{s}_t \leftarrow \text{ClampToROI}(\hat{s}_t, R)$
10:       **end if**
11:    **end for**
      // Stage 1: Associate with high-confidence detections
12:    $(\mathcal{T}_1, \mathcal{D}_1) \leftarrow \text{Associate}(\mathcal{T}, \mathcal{D}_{\text{high}})$
13:    $\mathcal{T}_{\text{remain}} \leftarrow \mathcal{T} \setminus \mathcal{T}_1$
14:    $\mathcal{D}_{\text{remain}} \leftarrow \mathcal{D}_{\text{high}} \setminus \mathcal{D}_1$
      // Stage 2: Associate with low-confidence detections
15:    $(\mathcal{T}_2, \mathcal{D}_2) \leftarrow \text{Associate}(\mathcal{T}_{\text{remain}}, \mathcal{D}_{\text{low}})$
16:    $\mathcal{T}_{\text{remain}} \leftarrow \mathcal{T}_{\text{remain}} \setminus \mathcal{T}_2$
17:    $\mathcal{T} \leftarrow \mathcal{T}_1 \cup \mathcal{T}_2$
      // Occlusion handling
18:    $\mathcal{T}_{\text{occ}} \leftarrow \{t \in \mathcal{T}_{\text{remain}} \mid \exists t' \in \mathcal{T}, \text{IoU}(t, t') \geq \text{IoU}_{\text{min}}\}$
19:    **for** each $t \in \mathcal{T}_{\text{occ}}$ **do**
20:       $t.\text{occluded} \leftarrow \text{true};$
21:       $\text{DampenVelocity}(t)$          ( equation 2)
22:       $\text{EnlargeBox}(t)$          ( equation 3)
23:    **end for**
24:    $\mathcal{T}_{\text{remain}} \leftarrow \mathcal{T}_{\text{remain}} \setminus \mathcal{T}_{\text{occ}}$
      // Deletion policy
25:    delete tracks in $\mathcal{T}_{\text{remain}}$
26:    delete $t \in \mathcal{T}_{\text{occ}}$ if $t.\text{occ} > T_{\text{occ}}$
27:    $\mathcal{T} \leftarrow \mathcal{T} \cup \mathcal{T}_{\text{occ}}$
      // Tracklet initialization
28:    $\mathcal{D}_{\text{init}} \leftarrow \{d \in \mathcal{D}_{\text{remain}} \mid \max_{t \in \mathcal{T}} \text{IoU}(d, t) < K_{\text{init}}\}$
29:    $\mathcal{T} \leftarrow \mathcal{T} \cup \text{InitializeTracks}(\mathcal{D}_{\text{init}})$
30: **end for**
31: **return** $\mathcal{T}$

---

```
        EnlargeBox(t)

    # Deletion / Initialization policies
    delete_spurious(self.tracks, self.t_occ)
    init_new_tracks(D2, self.k_init)

    return self.tracks
```

## C.2 ALGORITHM

Algorithm 1 summarizes the overall FastTracker pipeline. At each frame, detections are divided into high- and low-confidence groups and associated with existing tracklets in two stages. Environment-aware constraints refine predicted states, while occluded tracklets are stabilized through velocity

damping and bounding-box enlargement. Finally, tracklet initialization and deletion policies ensure spurious identities are suppressed and true targets are consistently recovered.

## D EXPERIMENT SETTINGS, EVALUATION, AND HYPERPARAMETERS

**Datasets.** For our experiments, we utilized four datasets: MOT16, MOT17, MOT20 and our introduced benchmark dataset. MOT16 and MOT17 include a wide range of pedestrian tracking scenarios with both static and moving cameras, where MOT17 further incorporates multiple detector outputs for robust evaluation. MOT20 presents more challenging scenes with extremely crowded environments and heavy occlusions, testing the limits of detection and tracking performance. In addition to these, our custom benchmark introduces even more extreme conditions, featuring massive pedestrian-car crowds, frequent and prolonged occlusions, and visually cluttered scenes. These properties result in significant overlaps between individuals, pushing beyond the visual complexity of the existing MOT datasets and offering a valuable testbed for evaluating the real-world robustness of tracking algorithms.

**Metrics.** We evaluate tracking performance using a combination of established metrics. These include the CLEAR metrics (Bernardin & Stiefelhagen, 2008)—such as MOTA, false positives (FP), false negatives (FN), and identity switches (IDs)—as well as IDF1 (Ristani et al., 2016) and the more recent HOTA metric (Luiten et al., 2021). While MOTA offers a general measure of tracking accuracy, IDF1 focuses on the quality of identity association. HOTA provides a balanced evaluation by jointly accounting for detection accuracy, association consistency, and localization precision.

**Implementation Details.** For detection, we adopt YOLOX due to its effective balance between speed and accuracy. The detector's classification and localization heads are trained following the best practices established in prior work (Du et al., 2023). At inference time, we apply a non-maximum suppression (NMS) threshold of 0.75. Tracklet association uses an IoU threshold of 0.5, and exponential moving average (EMA) smoothing is applied with a momentum coefficient $\alpha$ of 0.8. We set $\text{IoU}_{\min}$ to 0.7, the initialization overlap threshold $K_{init}$ to 0.8, and the occlusion tolerance window $T_{occ}$ to 30 frames. Detection confidence thresholds for classification are $\tau_{low} = 0.2$ and $\tau_{high} = 0.65$, while NSA Kalman is used for motion model. The region of interest $\mathcal{M}$ is user-configurable, and if direction information is provided, the tracker adjusts the trajectory estimation accordingly. We set $0.75 \leq \gamma_{\text{velo}} \leq 0.9$, $\delta_p, \delta_v \in \{3, 4, 5\}$ frames and $1.1 \leq \beta_{\text{enlarge}} \leq 1.2$ depending on the object class type.

In post-processing, we limit the maximum gap for Gaussian smoothing interpolation (GSP) to 20 frames. For global linking, we use the MARS ReID dataset (Spr, 2016) for pedestrian training and a vehicle ReID dataset sourced from GIAOTracker for vehicles. It is trained for 60 epochs using Adam optimizer with a cross-entropy loss function and a cosine annealing learning rate schedule. During inference, candidate associations are filtered using a temporal threshold of 15 frames and a spatial distance cap of 70 pixels. Only link scores exceeding 0.9 are accepted. All experiments were conducted on a system equipped with an NVIDIA RTX 4060 GPU with 8GB VRAM.

**Benchmark.** Our benchmark is annotated following the standard MOTChallenge format to ensure compatibility with existing multi-object tracking tools and evaluation protocols. Each annotation file includes frame index, object ID, bounding box coordinates, confidence, and other conventional MOT fields. The primary difference from the original MOTChallenge specification is that our dataset contains multiple object classes rather than only pedestrians. To accommodate this, we have a column that specifies the category label for each bounding box. Apart from this addition, our annotation structure remains aligned with the MOT format, allowing users to integrate the dataset with minimal modifications. The benchmark consists of 9 training videos, 1 validation video, and 2 test videos, covering diverse traffic conditions and occlusion scenarios.

## E COMPLEXITY ANALYSIS

In this section, we report the runtime complexity of all YOLOX variants used in our experiments, as well as the timing of our tracking pipeline. Our goal is to provide a clear picture of both detection and tracking costs, and to support our claim that the full system operates in real time on desktop GPUs and can also run efficiently on edge devices such as the Jetson AGX Orin.

## E.1 DETECTION RUNTIME

Table 12 summarizes the inference times and GFLOPs for all YOLOX variants evaluated in the main paper. Measurements were performed using PyTorch on an RTX 4060 GPU with an input size of $1088 \times 608$. Consistent with prior works, which consider frame rates above 24 FPS as real-time, all YOLOX variants satisfy this requirement.

Table 12: Runtime and GFLOPs of YOLOX variants (RTX 4060, input size $1088 \times 608$). Detection Time denotes the backbone/neck/head forward pass, Final Time includes NMS and post-processing, and FPS is computed as $1000/$Final Time.

| Model | Detection Time (ms) | Final Time (ms) | FPS ($\uparrow$) | GFLOPs |
|---|---|---|---|---|
| YOLOX-X | 35.7 | 40.0 | 25.0 | 455.3 |
| YOLOX-L | 33.1 | 37.4 | 26.7 | 251.3 |
| YOLOX-M | 30.9 | 35.6 | 28.4 | 118.7 |
| YOLOX-S | 28.5 | 32.8 | 30.5 | 43.0 |
| YOLOX-Tiny | 28.1 | 32.5 | 30.9 | 24.5 |
| YOLOX-Nano | 25.8 | 30.2 | 33.2 | 4.0 |

## E.2 TRACKING RUNTIME

For tracking complexity, we compare our tracker to ByteTrack under identical conditions. Averaged over all benchmarks, ByteTrack runs at approximately 4.1 ms per frame ($\sim 244$ FPS), while our tracker operates at 4.3–4.4 ms per frame ($\sim 233$ FPS). The slight overhead comes from adding occlusion detection, box enlargement, velocity damping, and refined track lifecycle management, which collectively improve re-identification and stability.

### E.2.1 EDGE DEPLOYMENT

In addition to the PyTorch implementation, we provide a full C++ deployment of our pipeline (detection + tracking), enabling users to run the system efficiently on NVIDIA edge platforms such as the Jetson AGX Orin (Table 13. We benchmarked our system using both Torch and TensorRT (C++) backends on the Orin, and our results confirm that the pipeline remains suitable for real-time or near–real-time usage in practical edge applications.

Table 13: Performance of YOLOX variants on Jetson AGX Orin using Torch and TensorRT (C++).

| Model | Jetson Orin (Torch) | Jetson Orin (TensorRT C++) |
|---|---|---|
| YOLOX-X | $\sim$13.0 FPS | $\sim$19.5 FPS |
| YOLOX-L | $\sim$14.0 FPS | $\sim$21.0 FPS |
| YOLOX-M | $\sim$15.0 FPS | $\sim$22.5 FPS |
| YOLOX-S | $\sim$16.3 FPS | $\sim$24.5 FPS |
| YOLOX-Tiny | $\sim$16.5 FPS | $\sim$24.8 FPS |
| YOLOX-Nano | $\sim$18.0 FPS | $\sim$27.0 FPS |

# F ADDITIONAL RESULTS

## F.1 DELETION AND INITIALIZATION POLICIES

Table 14 reports the effect of varying $K_{\text{init}}$ on tracking performance. The first row (✗) corresponds to disabling the initialization policy without ROI/Dir guidance and occlusion handling, meaning only the deletion rule is active; this results in consistently weaker performance across all datasets, confirming the importance of controlled initialization. When $K_{\text{init}}$ is enabled, we observe that performance remains stable across a broad range of values, with moderate thresholds (0.7–0.85) yielding the best balance of MOTA and HOTA. In particular, $K_{\text{init}} = 0.8$ achieves the strongest

results on DanceTrack, MOT20, and FastTrack, while slightly lower values (0.7–0.75) are favorable for MOT17. The inclusion of the extended DanceTrack benchmark further validates the generality of our design, showing clear improvements from 91.6/64.4 (MOTA/HOTA) without initialization to 92.8/65.4 with $K_{\text{init}} = 0.8$. Overall, these findings demonstrate that our approach is robust to the choice of $K_{\text{init}}$, with a broad plateau of effective settings across diverse benchmarks.

Table 14: Ablation on the effect of $K_{\text{init}}$ across different benchmarks.

| $K_{\text{init}}$ | MOT17-val | | MOT20-val | | FastTrack | | DanceTrack | |
|---|---|---|---|---|---|---|---|---|
| | MOTA ↑ | HOTA ↑ | MOTA ↑ | HOTA ↑ | MOTA ↑ | HOTA ↑ | MOTA ↑ | HOTA ↑ |
| ✗ | 79.4 | 63.5 | 74.5 | 62.8 | 60.1 | 57.2 | 91.6 | 64.4 |
| 0.60 | 79.6 | 63.6 | 74.8 | 62.9 | 60.5 | 56.9 | 91.9 | 64.8 |
| 0.65 | 79.7 | 63.8 | 74.8 | 63.0 | 60.5 | 56.9 | 92.2 | 65.3 |
| 0.70 | 79.9 | **64.1** | 75.0 | 63.1 | 60.7 | 57.8 | 92.6 | 65.5 |
| 0.75 | **80.0** | 64.0 | 75.0 | 63.2 | 60.8 | 57.8 | 92.7 | **65.6** |
| 0.80 | 79.9 | 64.0 | **75.2** | **63.3** | 60.9 | **58.0** | **92.8** | 65.4 |
| 0.85 | 79.6 | 63.8 | 75.1 | 63.3 | **61.0** | 58.0 | 92.7 | 65.3 |
| 0.90 | 79.4 | 63.6 | 75.0 | 63.2 | 60.9 | 57.8 | 92.5 | 65.1 |

## F.2 OCCLUSION HANDLING

Table 15 shows the influence of the occlusion threshold $\text{IoU}_{\text{min}}$ on tracking performance. The first row (✗) corresponds to disabling both initialization and deletion policies while also removing the occlusion module without ROI/Dir guidance, which results in a clear drop in accuracy across all datasets—for example, MOTA/HOTA fall to 59.2/56.8 on FastTrack and 89.1/62.2 on DanceTrack. Once occlusion handling is enabled (along with deletion/initialization policies), performance improves substantially, with intermediate thresholds (0.65–0.75) consistently achieving the best trade-off. In particular, $\text{IoU}_{\text{min}} = 0.70$ yields the strongest overall results, with peak HOTA on MOT17 and FastTrack and the highest MOTA/HOTA on the DanceTrack benchmark, while $\text{IoU}_{\text{min}} = 0.65$ and 0.75 offer comparable improvements on MOT20. Very low or very high thresholds degrade accuracy, confirming the importance of balanced occlusion sensitivity for stable identity preservation.

Table 15: Ablation on the effect of $\text{IoU}_{\text{min}}$ across different benchmarks.

| $\text{IoU}_{\text{min}}$ | MOT17-val | | MOT20-val | | FastTrack | | DanceTrack | |
|---|---|---|---|---|---|---|---|---|
| | MOTA ↑ | HOTA ↑ | MOTA ↑ | HOTA ↑ | MOTA ↑ | HOTA ↑ | MOTA ↑ | HOTA ↑ |
| ✗ | 79 | 63.1 | 74.1 | 62.1 | 59.2 | 56.8 | 89.1 | 62.2 |
| 0.60 | 80.2 | 65.0 | 76.5 | 63.2 | 62.8 | 60.0 | 92.7 | 65.3 |
| 0.65 | **80.5** | 65.1 | **76.9** | 63.4 | 63.1 | 60.4 | 93 | 65.7 |
| 0.70 | 80.4 | **65.2** | 76.7 | 64.4 | **63.4** | **60.7** | **93.4** | 65.9 |
| 0.75 | 80.3 | 65.2 | 76.8 | **64.5** | 63.2 | 60.6 | 93.2 | **66.1** |
| 0.80 | 80.1 | 64.9 | 76.4 | 64.2 | 62.9 | 60.4 | 93 | 65.8 |
| 0.85 | 79.8 | 64.6 | 76.3 | 64.0 | 62.7 | 60.0 | 92.9 | 65.3 |
| 0.90 | 79.3 | 64.3 | 76.0 | 63.5 | 61.9 | 59.4 | 92.6 | 65.1 |

Table 16 evaluates the impact of the maximum occlusion duration $T_{\text{occ}}$ before track deletion. The first row (✗) corresponds to disabling both initialization/deletion policies and the occlusion module without ROI/Dir guidance. Once occlusion tolerance is introduced (along with deletion/initialization policies), moderate values (25–35 frames) provide the best balance across benchmarks. In particular, $T_{\text{occ}} = 30$ achieves the highest overall HOTA on MOT17, MOT20, and FastTrack, while $T_{\text{occ}} = 35$ yields slightly stronger MOTA on MOT17 and the best MOTA on DanceTrack. Very short thresholds remove tracks too aggressively, while overly long ones retain stale tracklets, confirming that carefully tuning $T_{\text{occ}}$ is essential for maintaining robust identity association.

Table 17 analyzes the impact of the enlargement factor $\beta_{\text{enlarge}}$ used to expand bounding boxes during occlusion. Once occlusion enlargement is applied (along with deletion/initialization policies),

Table 16: Ablation on the effect of $T_{\text{occ}}$ across different benchmarks.

| $T_{\text{occ}}$ | MOT17-val | | MOT20-val | | FastTrack | | DanceTrack | |
|---|---|---|---|---|---|---|---|---|
| | MOTA ↑ | HOTA ↑ | MOTA ↑ | HOTA ↑ | MOTA ↑ | HOTA ↑ | MOTA ↑ | HOTA ↑ |
| ✗ | 79 | 63.1 | 74.1 | 62.1 | 59.2 | 56.8 | 89.1 | 62.2 |
| 20 | 80.1 | 64.8 | 76.4 | 64.0 | 62.4 | 59.7 | 92.8 | 64.9 |
| 25 | 80.3 | 65.0 | **76.7** | 64.2 | 63.2 | 60.4 | 93.3 | 65.6 |
| 30 | 80.4 | **65.2** | 76.7 | **64.4** | **63.4** | **60.7** | 93.4 | **65.9** |
| 35 | **80.5** | 65.1 | 76.6 | 64.3 | 63.4 | 60.5 | **93.6** | 65.7 |
| 40 | 80.3 | 65.0 | 76.4 | 64.3 | 63.0 | 60.3 | 93.1 | 65.3 |
| 45 | 80.2 | 64.8 | 76.4 | 64.1 | 62.8 | 59.9 | 92.9 | 65 |
| 50 | 80.0 | 64.7 | 76.3 | 64.0 | 62.0 | 59.6 | 92.5 | 64.6 |

moderate factors around 1.15–1.20 provide the most consistent gains. In particular, $\beta_{\text{enlarge}} = 1.15$ achieves the best overall balance with peak HOTA on MOT17 and FastTrack, while $\beta_{\text{enlarge}} = 1.20$ slightly improves MOTA on MOT17 and achieves the strongest MOTA on DanceTrack. Both very small and overly large enlargement factors degrade performance, confirming that controlled bounding box expansion is critical for recovering occluded targets without adding false positives.

Table 17: Ablation on the effect of $\beta_{\text{enlarge}}$ across different benchmarks.

| $\beta_{\text{enlarge}}$ | MOT17-val | | MOT20-val | | FastTrack | | DanceTrack | |
|---|---|---|---|---|---|---|---|---|
| | MOTA ↑ | HOTA ↑ | MOTA ↑ | HOTA ↑ | MOTA ↑ | HOTA ↑ | MOTA ↑ | HOTA ↑ |
| ✗ | 79 | 63.1 | 74.1 | 62.1 | 59.2 | 56.8 | 89.1 | 62.2 |
| 1.05 | 80.0 | 64.7 | 76.3 | 64.2 | 62.8 | 59.9 | 92.7 | 65.2 |
| 1.10 | 80.3 | 65.0 | 76.5 | **64.5** | 63.1 | 60.5 | 93.4 | **66** |
| 1.15 | 80.4 | 65.2 | **76.7** | 64.4 | **63.4** | **60.7** | 93.4 | 65.9 |
| 1.20 | **80.6** | **65.3** | 76.6 | 64.3 | 63.4 | 60.5 | **93.7** | 65.7 |
| 1.25 | 80.3 | 65.0 | 76.4 | 64.3 | 63.0 | 60.3 | 93.2 | 65.3 |
| 1.30 | 80.2 | 64.8 | 76.4 | 64.1 | 62.8 | 59.9 | 92.8 | 64.9 |

Table 18 studies the impact of the velocity damping factor $\gamma_{\text{velo}}$ used to stabilize tracklets during occlusion. Without damping (✗), performance drops significantly across all benchmarks, highlighting the importance of controlling velocity updates. Moderate damping values between 0.85–0.90 yield the strongest and most consistent improvements. In particular, $\gamma_{\text{velo}} = 0.85$ achieves the best overall balance with peak MOTA and HOTA on MOT17, MOT20, and FastTrack, while $\gamma_{\text{velo}} = 0.90$ further boosts HOTA on MOT17 and achieves the highest MOTA on DanceTrack. Both weaker damping (e.g., 0.75) and excessive damping degrade results, confirming that carefully tuned velocity suppression is crucial for recovering stable trajectories under occlusion.

Table 18: Ablation on the effect of $\gamma_{\text{velo}}$ across different benchmarks.

| $\gamma_{\text{velo}}$ | MOT17-val | | MOT20-val | | FastTrack | | DanceTrack | |
|---|---|---|---|---|---|---|---|---|
| | MOTA ↑ | HOTA ↑ | MOTA ↑ | HOTA ↑ | MOTA ↑ | HOTA ↑ | MOTA ↑ | HOTA ↑ |
| ✗ | 79 | 63.1 | 74.1 | 62.1 | 59.2 | 56.8 | 89.1 | 62.2 |
| 0.75 | 79.9 | 64.4 | 75.9 | 63.8 | 61.9 | 59.6 | 92.5 | 64.7 |
| 0.80 | 80.2 | 64.9 | 76.5 | 64.1 | 62.5 | 60.2 | 93 | 65.4 |
| 0.85 | **80.4** | 65.2 | **76.7** | 64.4 | **63.4** | **60.7** | 93.4 | **65.9** |
| 0.90 | 80.3 | **65.3** | 76.7 | **64.5** | 63.1 | 60.4 | **93.8** | 65.8 |

Table 19 examines the influence of the reset factors $(\delta_p, \delta_v)$ used to restore position and velocity after occlusion detection. The first row (✗, ✗) removes all occlusion-reset logic without ROI/Dir guidance, leading to the weakest results across benchmarks (e.g., only 59.2/56.8 MOTA/HOTA on FastTrack and 89.1/62.2 on DanceTrack). Once reset is enabled (along with deletion/initialization policies),

moderate offsets in the range of 4–5 consistently yield the strongest performance. In particular, $(\delta_p{=}4, \delta_v{=}5)$ achieves the best overall balance across datasets, with especially strong gains on MOT20, FastTrack, and DanceTrack. Larger settings such as $(5,5)$ slightly improve HOTA on MOT17 and DanceTrack but show diminishing returns. Very small offsets (e.g., $(3,3)$) under-correct trajectories, while overly aggressive ones risk drift, confirming the need for careful joint tuning of both reset factors.

Table 19: Ablation on the effect of $(\delta_p, \delta_v)$ across different benchmarks.

| $\delta_p$ | $\delta_v$ | MOT17-val | | MOT20-val | | FastTrack | | DanceTrack | |
|---|---|---|---|---|---|---|---|---|---|
| | | MOTA ↑ | HOTA ↑ | MOTA ↑ | HOTA ↑ | MOTA ↑ | HOTA ↑ | MOTA ↑ | HOTA ↑ |
| ✗ | ✗ | 79 | 63.1 | 74.1 | 62.1 | 59.2 | 56.8 | 89.1 | 62.2 |
| 3 | 3 | 80.1 | 64.4 | 76.0 | 64.3 | 62.9 | 60.0 | 92.8 | 65.2 |
| 3 | 4 | 80.2 | 64.6 | 76.1 | 64.3 | 63.0 | 60.0 | 92.9 | 65.3 |
| 3 | 5 | 80.2 | 64.7 | 76.3 | 64.4 | 63.2 | 60.3 | 93.1 | 65.5 |
| 4 | 3 | 80.3 | 65.1 | 76.5 | 64.2 | 63.1 | 60.5 | 93.1 | 65.5 |
| 4 | 4 | **80.4** | 65.2 | 76.5 | 64.2 | 63.4 | 60.4 | 93.3 | 65.7 |
| 4 | 5 | 80.4 | 65.2 | 76.7 | **64.4** | **63.4** | **60.7** | **93.4** | 65.9 |
| 5 | 3 | 80.0 | 65.0 | 76.4 | 64.2 | 63.1 | 60.3 | 92.9 | 65.8 |
| 5 | 4 | 80.0 | 65.2 | 76.7 | 64.4 | 63.1 | 60.6 | 93 | 65.9 |
| 5 | 5 | 80.2 | **65.4** | **76.8** | 64.4 | 63.3 | 60.6 | 93.1 | **66.1** |

## F.3 MOT16

Table 20 presents the official MOT16 benchmark results. Our method, FastTracker, achieves the highest scores, with MOTA of 79.1 and HOTA of 66.0, surpassing recent state-of-the-art approaches. Compared to FeatureSORT (MOTA 77.9, HOTA 62.8), we improve by +1.2 MOTA and +3.2 HOTA. Similarly, against StrongSORT (MOTA 77.8, HOTA 63.8), we see gains of +1.3 MOTA and +2.2 HOTA. Notably, we also achieve the lowest number of ID switches (290), demonstrating superior identity preservation throughout the sequence.

Table 20: MOT16: Performance comparison with SOTA

| Method | MOTA↑ | HOTA↑ | IDF1↑ | FP↓ | FN↓ | IDs↓ |
|---|---|---|---|---|---|---|
| QDTrack (Pang et al., 2021) | 69.8 | 54.5 | 67.1 | 9861 | 44050 | 1097 |
| TraDes (Wu et al., 2021) | 70.1 | 53.2 | 64.7 | **8091** | 45210 | 1144 |
| CSTrack (Liang et al., 2022b) | 75.6 | 59.8 | 73.3 | 9646 | 33777 | 1121 |
| GSDT (Wang et al., 2021b) | 74.5 | 56.6 | 68.1 | 8913 | 36428 | 1229 |
| RelationTrack (Yu et al., 2022) | 75.6 | 61.7 | 75.8 | 9786 | 34214 | 448 |
| OMC (Liang et al., 2022a) | 76.4 | 62.9 | 74.1 | 10821 | 31044 | 1087 |
| CorrTracker (Wang et al., 2021a) | 76.6 | 61.0 | 74.3 | 10860 | 30756 | 979 |
| SGT (Hyun et al.) | 76.8 | 61.2 | 73.5 | 10695 | 30394 | 1276 |
| FairMOT (Zhang et al., 2021) | 74.9 | 58.3 | 72.8 | 9952 | 38451 | 1074 |
| StrongSORT (Du et al., 2023) | 77.8 | 63.8 | 78.3 | 11254 | 32584 | 1538 |
| FeatureSORT (Hashempoor et al., 2024) | 77.9 | 62.8 | 76.3 | 14827 | **26877** | 597 |
| FastTracker | **79.1** | **66.0** | **81.0** | 8785 | 29028 | **290** |

## F.4 BDD100K

Fasttracker achieves the strongest performance on BDD100K among all compared trackers. As shown in Table 21, it reaches 43.8 mMOTA and 56.2 mIDF1, outperforming both traditional trackers (e.g., SORT, DeepSORT, MOTDT) and more advanced deep multi-object tracking frameworks such as QDTrack, TETer, Unicorn, and MOTR. Notably, it also surpasses BYTE (with ReID) by a clear margin, demonstrating the effectiveness of our occlusion-handling strategy and lightweight motion stabilization modules in challenging real-world driving scenes.

Table 21: Comparison of tracking performance on the BDD100K benchmark. Fasttracker achieves the highest mMOTA and mIDF1 among all evaluated methods.

| Method | mMOTA (↑) | mIDF1 (↑) |
|---|---|---|
| SORT | 30.9 | 41.3 |
| DeepSORT | 24.5 | 38.2 |
| MOTDT | 26.7 | 39.8 |
| BYTE (with ReID) | 40.1 | 52.8 |
| QDTrack (Pang et al., 2021) | 36.6 | 50.8 |
| TETer (Li et al., 2022) | 39.1 | 53.3 |
| Unicorn (Yan et al., 2022) | 41.2 | 54.0 |
| MOTR (Zeng et al., 2022) | 35.5 | 48.2 |
| **Fasttracker (ours)** | **43.8** | **56.2** |

## F.5 VISUALIZATIONS

To have visualizations, we provide qualitative results in Figures 10–12, illustrating how our proposed modules recover targets under challenging occlusions. Figures 10 and 11 show cases where ROI and direction constraints (`ProjectToCone`, `ClampToROI`) prevent trajectories from drifting outside valid regions and enable successful recovery of occluded targets, which the baseline fails to achieve. Figure 12 highlights the effect of `DampenVelocity` and `EnlargeBox`, where moderating motion and enlarging the bounding box allow re-identification after occlusion. Together, these examples demonstrate the practical effectiveness of our proposed modules in maintaining identity consistency. Although Fasttracker successfully handles most occlusion cases, there are still scenarios where it can fail if the situation becomes extremely complex. For example, when an object is fully occluded and its direction or motion changes during the occlusion. An example of such a failure case is shown in Figure 13. It is worth noting, however, that while Fasttracker fails in these rare edge cases, all baseline methods fail not only here but also in many simpler scenarios that Fasttracker handles reliably.

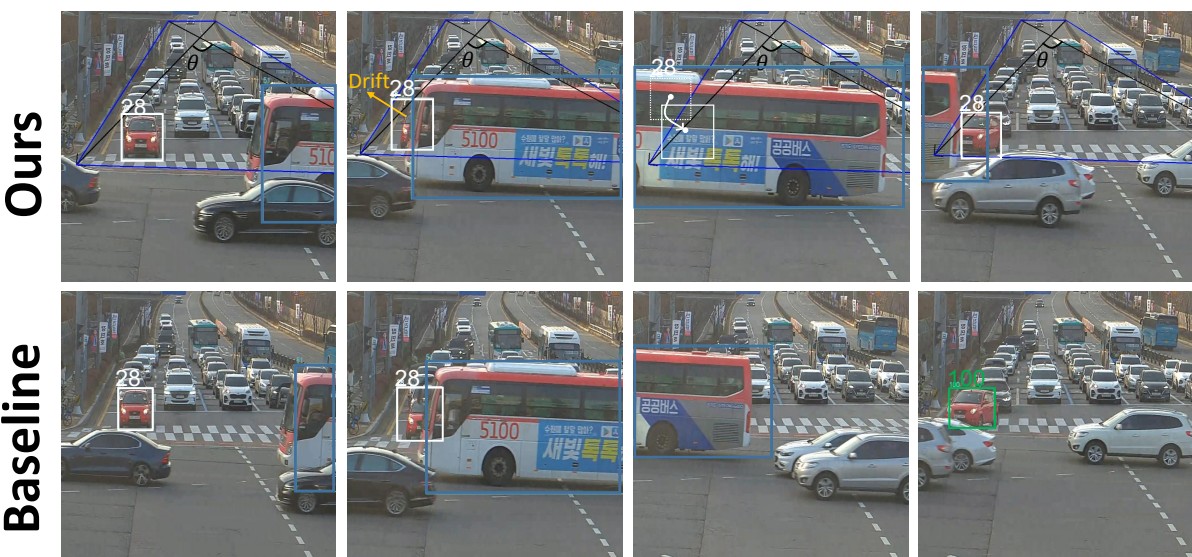

Figure 10: Illustration of ROI and direction constraints. **Top row:** In frame 2, target ID 28 is occluded by ID 51, causing its trajectory to drift outside the valid region. In frame 3, the direction limitation and ROI constraint modules (`ProjectToCone`, `ClampToROI`) correct the trajectory back into the allowable region. By frame 4, ID 28 is successfully recovered. **Bottom row:** The baseline result fails to recover the ID.

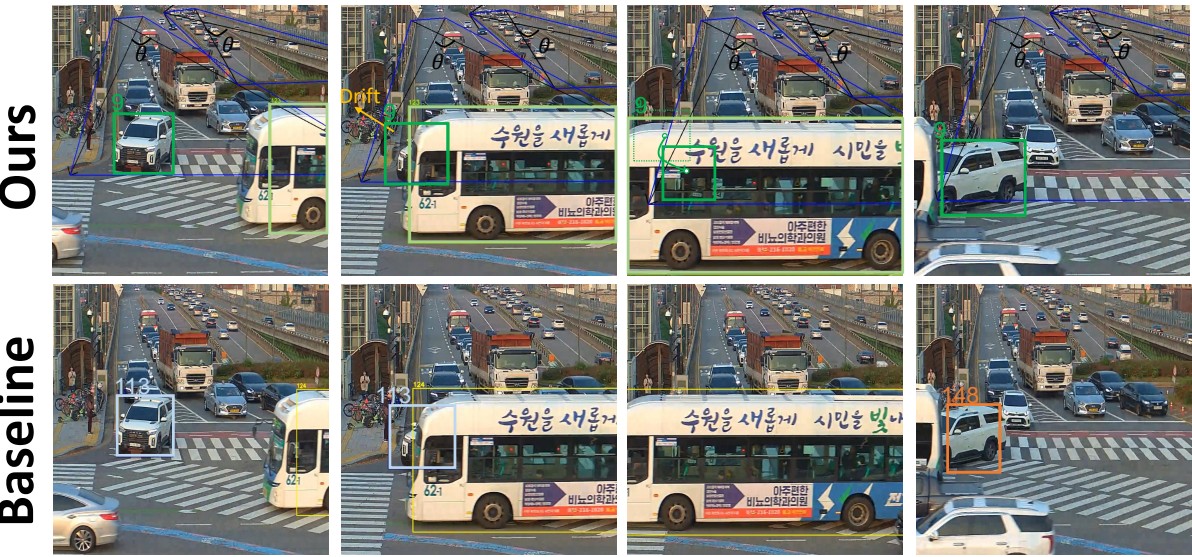

Figure 11: Illustration of ROI and direction constraints. **Top row:** In frame 2, target ID 9 is occluded by ID 123, causing its trajectory to drift outside the valid region. In frame 3, the direction limitation and ROI constraint modules (`ProjectToCone`, `ClampToROI`) correct the trajectory back into the allowable region. By frame 4, ID 9 is successfully recovered. **Bottom row:** The baseline result fails to recover the ID.

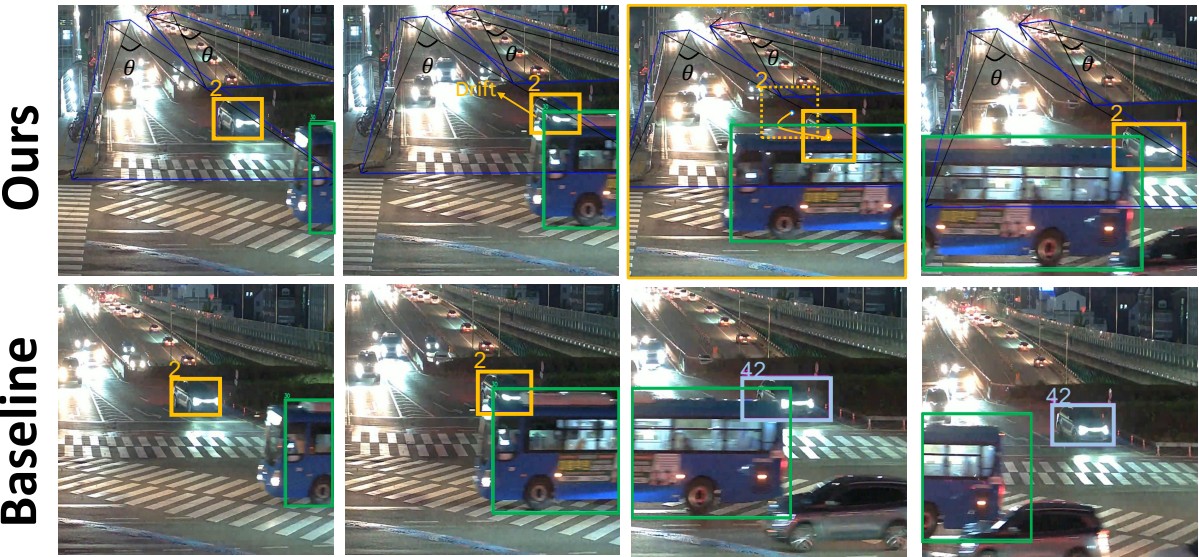

Figure 12: Illustration of `EnlargeBox` and `DampenVelocity` (including reset of positions and velocities). **Top row:** In frame 2, target ID 2 is occluded by ID 30, causing its trajectory to drift. In frame 3, velocity damping and box enlargement stabilize the trajectory and keep it within a plausible region for re-identification. By frame 4, ID 2 is successfully recovered. **Bottom row:** The baseline result fails to recover the ID.

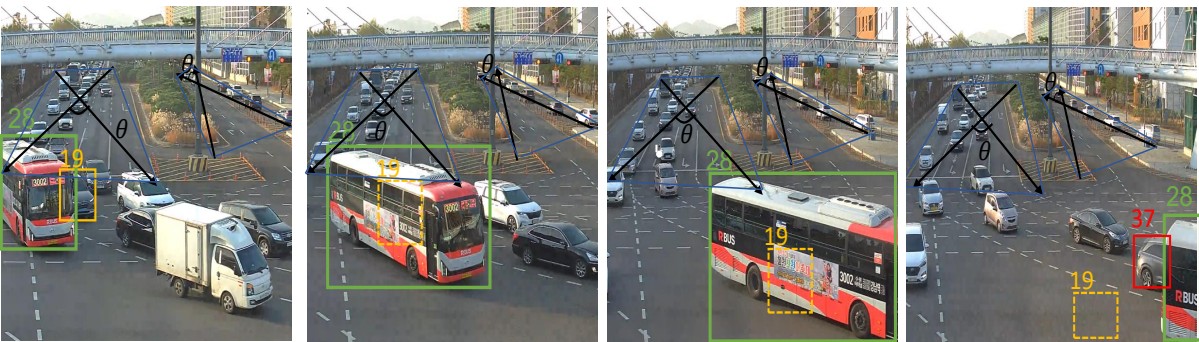

Figure 13: The sedan (ID 19) follows a curved trajectory and becomes heavily occluded by the bus (ID 28), causing the tracker to lose consistent visual cues. Although the tracker keeps propagating ID 19 during the occlusion, once the sedan becomes visible again (fourth frame) it cannot be matched to any existing tracklet, resulting in a new ID being assigned (ID 37). Since ID 19 remains unmatched for several consecutive non-occluded frames, it is eventually removed from the active tracklets.

