# OpenReview forum: "FastTracker: Real-Time and Accurate Visual Tracking"
_ICLR.cc/2026/Conference — Submitted to ICLR 2026_

### Official Review · Reviewer_Rd4h · 2025-10-27

**Soundness:** 3
**Presentation:** 3
**Contribution:** 3
**Rating:** 6
**Confidence:** 5

**Summary:**

The paper proposes FastTracker, a lightweight, real-time multi-object tracking framework. Its core contributions include: occlusion handling without ReID via DampenVelocity and EnlargeBox，and trajectory regularization using environment priors defined by a direction cone and a quadrilateral ROI. In addition, the authors introduce FastTrack, an internal CCTV multi-class traffic benchmark.

**Strengths:**

1、Originality: Explicitly encode the scene prior with ProjectToCone to correct the prediction before association. Use center proximity for occlusion detection and stabilize trajectories with DampenVelocity and EnlargeBox.
2、Quality: The method is well designed, the modules are interrelated, and the experimental pipeline is clear, with ablation and comparative experiments provided to demonstrate the framework’s performance.
3、Clarity: The flowcharts, visual examples, and algorithmic steps are easy to follow, and the experimental results are comprehensive.
4、Significance: By modularizing the various components, the paper provides a reusable, comparable baseline for subsequent Kalman-based trackers, and provides a dataset for complex scenarios that meaningfully advances engineering evaluation in MOT.

**Weaknesses:**

1、The association backbone follows ByteTrack’s two-stage matching scheme; the method’s novelty lies primarily in the pre-association constraints and occlusion heuristics rather than in the association paradigm itself.
2、Dependence on priors: The ROI/direction constraints must be predefined manually, and the ROI is restricted to a quadrilateral. This strong reliance on scene priors limits deployment and generalization.
3、Implementation details: Although the paper states that a Kalman filter is used, it does not fully specify the motion model, including the definition of the state vector and the settings for process noise and measurement noise.

**Questions:**

1、Will the FastTrack dataset be made publicly available? Please describe the privacy handling, data collection, and the compliance policies and procedures for external release.
2、How robust is the method when scene priors are missing or mis-registered? Does performance degrade substantially in such cases?
3、Please provide the complete mathematical specification of the NSA Kalman filter and the default parameter settings for categories such as pedestrians and bicycles.

---

> ### Author Response · Authors · 2025-11-24
> **Reply to reviewer Rd4h**
>
> Thank you for your useful comments and positive feedback about our draft. We address your Qs one by one in the following:
>
> ---
>
> **Q**: *association backbone follows ByteTrack ...?*
>
> **A**:
> In MOT, improvements come from different directions: some works focus on stronger ReID features or new architectures, some touch the assignment/matching procedure, and some introduce environment-aware cues (e.g., SparseTrack, PD-SORT) with management policies to improve stability in difficult scenes. Our approach belongs to the last group.
> Our main contribution is not to redesign the association paradigm, but to show that simple, lightweight constraints and occlusion-handling rules, along with del/init policies, can improve identity stability without relying on heavy appearance models. In addition, we place emphasis on practical deployment of our work as another contribution: the full system runs efficiently across different platforms, including edge devices, and we provide additional results for this purpose (general response above). We think this combination of robust heuristics, strong deployment characteristics, and a dedicated benchmark makes the contribution meaningful even though the core association framework is conventional.
>
> ---
>
> **Q**: *Automatic priors? reliance on quadrilateral ROI...?*
>
> **A**:
> We agree that relying on a manually defined quadrilateral ROI is a limitation, but this is not a fundamental one. A practical way to handle more complex scenes is to split the area into several smaller quadrilateral sub-regions and treat each of them separately. For instance, an intersection may contain eight directional lanes (two per side: north, east, south, west) plus a central turning area. Each lane can be modeled as its own quadrilateral ROI, while the central shared zone can simply be treated as a non-ROI region where only occlusion handling and track-lifecycle rules apply. If automatic ROI priors are available, these sub-regions can even be generated automatically, making the system flexible for more complicated layouts.
>
> For the automatic priors, it is completely feasible to employ it. We did an initial test to check the performance, where we used an ENet model trained on Cityscapes to produce pixel-wise scene labels and extract the drivable region using a simple contour approximation. This gives a reasonable polygonal ROI sometimes slightly imperfect, but still clearly better than having no ROI at all. More advanced segmentation networks can be plugged in whenever better scene understanding is needed.
>
> ---
>
> **Q**: *dataset...?*
>
> **A**:
> Yes, Sure. The readers can use dataset without any limit and further details are added in appendix about annotation (please kindly refer to point 1 in general response above).
>
> ---
>
> **Q**: *robustness of the method when scene priors are missing or mis-registered...?*
>
> **A**:
> Our method stays fairly robust even when the scene priors are not perfect. In all FastTracker benchmark scenarios, the ROI and direction-angle settings worked consistently well, and the system showed good tolerance to noise or small mistakes in the ROI configuration. Small deviations in $\theta$ or misplaced ROI points do not noticeably affect the results because of the confidence threshold (we allow a few degrees of deviation from theoretical $\theta$ that we derive). We also tested ROIs shifted by a few meters around the actual drivable area, and the performance only dropped slightly, still clearly better than running the tracker with no ROI at all. Only very large mistakes, such as reversing the road direction by 180 degrees, can hurt performance. In practical setups this kind of extreme error is unlikely, so overall the method handles real-world imperfections quite well.
>
> ---
>
> **Q**: *complete mathematical specification of the NSA Kalman filter...?*
>
> **A**:
> All requested details have been added to Appendix B. We now include the full mathematical formulation of both the standard Kalman motion model and its NSA extension, along with the class-aware modification used in our tracker. As explained in the appendix, the only category-dependent parameter is the time-step factor $dt$ inside the transition matrix $\mathbf{G}_{\text{cls}}$ (in Kalman formulation), which controls how fast an object is allowed to move between frames. Faster categories use larger $dt$ values, while slower ones use smaller values, enabling more realistic motion prediction in occlusions or temporary detection dropouts. The specific values used in our system are: person ($dt=0.20$), bike ($dt=0.35$), pm/scooter ($dt=0.40$), motorbike ($dt=0.55$), car ($dt=1.00$), truck small ($dt=0.85$), truck big ($dt=0.80$), bus small ($dt=0.75$), and bus big ($dt=0.70$). All are involved in table 11 of appendix in revised draft now.

---

### Official Review · Reviewer_X9FY · 2025-10-27

**Soundness:** 3
**Presentation:** 2
**Contribution:** 2
**Rating:** 4
**Confidence:** 4

**Summary:**

FastTracker proposes a lightweight, general multi-object tracking (MOT) framework for vehicle-rich traffic scenes. Building on ByteTrack’s two-stage association, it replaces deep re-identification with motion and spatial cues. Two key modules are introduced: (1) an occlusion-aware re-ID using geometric coverage and velocity dampening for ID continuity, and (2) a road-structure-aware refinement leveraging manually defined ROIs and lane-direction constraints to enforce plausible trajectories. The authors also release the FastTrack dataset (9 classes, 12 scenes, 800K annotations) for vehicle-centric tracking. It achieves state-of-the-art results on MOT17 (HOTA 66.4), MOT20 (65.7), DanceTrack, and the new dataset, with notably fewer ID switches. Ablation studies show consistent gains from each module.

**Strengths:**

- Comprehensive Experiments: The authors evaluate on a wide range of benchmarks (MOT16/17/20, DanceTrack, and FastTrack) with detailed ablation studies (Tables 2–5) verifying each module’s impact. The improvements in key metrics (MOTA, HOTA, IDF1) and especially the reduction in ID switches (e.g. lowest IDs on MOT17/20) are convincingly shown.

- Strong Empirical Results: FastTracker achieves state-of-the-art or competitive performance on multiple datasets. For instance, it reaches HOTA 66.4 on MOT17 and 65.7 on MOT20 (test sets), and outperforms all baselines on DanceTrack (MOTA 93.4, HOTA 65.9). These results indicate a high practical impact.

- New Benchmark Dataset: The introduction of the FastTrack dataset (800K frames, 9 classes, diverse traffic scenes) fills a notable gap. The paper provides dataset statistics and sample frames, and demonstrates that existing trackers perform significantly worse on it, highlighting its challenge. Making this dataset available would be a great resource for the community.

**Weaknesses:**

- Manual ROI/Direction Constraints: A key limitation is the reliance on manually defined polygonal ROIs and fixed cone directions for scene priors. As acknowledged by the authors, this is labor-intensive and may not generalize to complex or evolving environments (e.g. intersections, roundabouts). The current system only supports quadrilateral regions, limiting flexibility. This reliance diminishes the novelty and practicality of the road-structure module.

- Lack of Runtime Analysis: The claim of real-time operation is not quantitatively supported. The paper reports performance with lighter detectors but does not give actual frame rates or hardware details. It is unclear whether the occlusion and ROI modules introduce any latency, or how the system performs on typical edge devices.

- Many parts of the system (velocity dampening factor, coverage thresholds, direction-angle limits) are hand-designed. While ablations show they work, it is not clear how sensitive the system is to these hyperparameters. In some cases, learned re-ID features might be more robust to varied conditions than the proposed heuristics.

**Questions:**

- ROI/Direction Robustness: How sensitive is FastTracker to inaccuracies in the manually defined ROI or direction cones? If the annotated road boundaries are slightly off, does performance degrade significantly?

- Automatic Priors: Do the authors plan to incorporate automatic scene understanding (e.g. segmentation of roads/lanes) to generate the ROI and direction constraints? Have any preliminary tests been done in this direction?

- Runtime Performance: Can the authors provide empirical runtime measurements (e.g. frames per second) for FastTracker on a typical hardware setup, both with the heavy (YOLOX-L) and lightweight detectors (YOLOX-Nano)?

- Failure Cases: Are there particular scenarios where FastTracker fails (e.g. very long occlusions, heavy clutter)? Can you provide qualitative examples of its limitations?

- Dataset Release: Will the FastTrack benchmark and annotations be made publicly available, and under what license?

---

> ### Author Response · Authors · 2025-11-24
> **Reply to reviewer X9FY**
>
> Thank you so much for all of the time you invest and comments provided for our paper. We reply to the Qs one by one:
>
> ---
>
> **Q**: *ROI/Direction Robustness and heuristic features ...?*
>
> **A**:
> Our ROI and direction-angle settings were tested in all scenarios of the FastTracker benchmark and worked consistently well. The system is quite tolerant to noise or mistakes in the ROI setup. For example, small errors in $\theta$ or slightly inaccurate ROI points do not change the results thanks to our confidence threshold. Only very large mistakes like flipping the road direction by 180 degrees can hurt the performance. In normal setups this is unlikely to happen, and overall the method handles real-world imperfections well.
>
> ---
>
> **Q**: *re-ID features might be more robust ...?*
>
> **A**:
> Regarding the robustness of re-ID features, we generally agree with the reviewer. However, depending on the re-ID network design, the extracted features may become too generic especially for vehicles that share similar appearances (e.g., color or shape so most of cars share similar features). On the other hand, deploying very strong re-ID networks often comes at the cost of real-time speed. Our focus has been to strengthen lightweight parts of the tracking pipeline to preserve real-time performance across a broad range of devices, even edges (please see the complexity part comment in our general response above).
>
> ---
>
> **Q**: *quadrilateral constraint ...?*
>
> **A**:
> We agree that this a current limitation, but at the same time, this constraint is not fundamental. A straightforward extension is to split a complex area into several simpler quadrilateral sub-regions and treat each one independently. For example, in an intersection you often have eight directional lanes (two per each side: north, east, south, west) plus a central shared area where vehicles can turn. Each of those directional lanes can be modeled as its own quadrilateral ROI, while the central shared zone can simply be treated as a non-ROI area where only occlusion handling and track lifecycle policies are applied (visualized in the figures in appendix).
> If automatic ROI priors are available, these sub-regions can even be generated automatically, making the approach flexible enough for more complicated layouts.
>
> ---
>
> **Q**: *Automatic priors ...?*
>
> **A**:
> One important note regarding ROI detection is that it does not impact the real-time performance of our method, since it only needs to be performed once at the beginning (e.g., during the setup stage) or periodically, not per frame. We did an initial test and used an ENet segmentation model trained on the Cityscapes dataset to get pixel-wise labels of the scene (e.g., road, sidewalk). From the predicted road mask, we extracted the drivable area using simple contour approximation and used it as a polygonal ROI. This produces a reasonable baseline ROI: it may have inaccuracies (because of not strong ENet baseline, but easy-to-employ for users), but it still performs better than using no ROI at all. More advanced segmentation models can easily be plugged in if higher-quality scene understanding is needed.
>
> ---
>
>
> **Q**: *Runtime Performance ...?*
>
> **A**: Thanks for suggesting. Yes, added in 'RunTime-Complexity' response above. Besides, they are also in the revised draft (Appendix E).
>
> ---
>
>
> **Q**: *Failure Cases ...*
>
> **A**:
> There are many complex occlusion scenarios for example:
> (i) long static occlusions, such as a vehicle fully hidden behind a bus at a red light;
> (ii) double-layer occlusions, where a vehicle is first occluded by another car, and then both are occluded by a larger vehicle like a bus;
> (iii) dynamic long occlusions, for instance when a car is hidden by a turning bus while both are moving and changing direction (basically the movement patterns change).
> FastTracker handles many typical occlusion cases well, but under extremely complex and prolonged occlusions with direction changes (like some cases in (iii)), we observed occasional ID switches or losses. For example, when both an occluder and the occluded object change direction simultaneously while fully occluded for a long time, re-identification becomes more difficult (we added this visualization in the rebuttal draft figure 13 for the reviewer with an explanation about it in F.5) .
> It is worth noting that baseline trackers fail in all cases even in many simpler short occlusion scenarios where FastTracker still performs reliably.
>
> ---
>
>
> **Q**: *Dataset Release ...*
>
> **A**: Yes, users can use dataset without any limit and further details are added in appendix about annotation (kindly refer to answer 1 in general response above).
>
> ---

---

### Official Review · Reviewer_xSDA · 2025-11-01

**Soundness:** 2
**Presentation:** 3
**Contribution:** 2
**Rating:** 4
**Confidence:** 3

**Summary:**

The paper proposes FastTracker, a lightweight, motion-centric online MOT framework designed for multi-class tracking in complex urban scenes, with particular emphasis on vehicles. The method builds upon a two-stage association strategy (high- then low-confidence detections) and augments it with: (i) an occlusion-aware module that stabilizes track states without CNN-based ReID by damping velocity and enlarging boxes during occlusions, and (ii) environment-aware constraints based on road geometry and directional priors (ProjectToCone, ClampToROI). The authors also introduce a CCTV-based benchmark (FastTrack) with diverse traffic scenarios and claim state-of-the-art results on MOT16/17/20 and DanceTrack while remaining real-time and resource efficient.

**Strengths:**

1. Clear problem motivation: addresses generalization beyond pedestrian tracking and the need for multi-class vehicle-centric tracking under occlusions and complex layouts.
2. Practical, lightweight design: avoids deep appearance models in the online pipeline; relies on motion, geometry, and simple heuristics that are attractive for real-time deployments.
3. Environment-aware modeling: novel use of region semantics and directional constraints to limit drift and enforce plausible motion without heavy learning modules.

**Weaknesses:**

1. Clarity and correctness of some definitions:
The “center-proximity score CP” is described as computed via IoU, which is conceptually inconsistent (center-proximity is not IoU). A precise definition is missing.
2. Occlusion handling design details:
Marking occlusion based on overlap with other active tracklets via a single threshold may conflate crowding with occlusion and induce false occlusion states.
3. Dataset details and release:
The FastTrack dataset has only 12 videos (albeit very dense). More details are needed: annotation protocol, quality control, train/val/test splits, licensing, and release plan. Without public release, the dataset’s impact is limited.

**Questions:**

1. Please precisely define CP(t, t′) and how occlusion is decided, including edge cases in crowded scenes. Is CP IoU, or a center-distance metric?
2. How are occluded tracks re-associated upon reappearance if they are excluded from association? What gating, timing, and matching logic ensure ID continuity?
3. Do you output predicted boxes during occlusion? If yes, how do enlarged boxes affect FP/FN and HOTA?

---

> ### Author Response · Authors · 2025-11-23
> **Reply to reviewer xSDA (1)**
>
> **Q**: *Clarity and CP ...*
>
> **A**:
> Thank you for the comment and provided useful details to us. We agree the term *center-proximity score* was misleading, as we are in fact using an IoU-based overlap check between tracklets to determine occlusion. We have revised the description to replace the CP term with the correct IoU-based formulation: *we compute $\mathrm{IoU}(t, t')$ and assess whether the spatial overlap exceeds a defined threshold to consider an occlusion...* (now revised in lines 208-210)
>
> ---
>
> **Q**: *Occlusion handling design details ... Edge cases ...*
>
> **A**:
> In crowded scenes, multiple partial overlaps between an unmatched tracklet and several active ones are indeed possible. However, to ensure robust occlusion detection, we compute the IoU between each unmatched tracklet and all currently active ones and only mark it as occluded if at least one IoU exceeds a defined threshold. This avoids false positives because of minor overlap. In practice, we ablated two strategies: 1- using the maximum single overlap (i.e., the highest IoU with any active tracklet), and 2- summing all IoUs from overlapping active tracklets. We observed that the first approach (focusing on the major occluder) produced more stable results. This is likely because true multiple-object occlusion happens in very few frames, and summing small overlaps increases noise sensitivity.
>
> Regarding edge cases in occlusion detection, we handle several types of non-typical scenarios in our framework. For long occlusions, tracklets are allowed to remain in the system with damped dynamics until they either reappear (and are reassociated) or exceed the occlusion timeout threshold and deleted. For short occlusions or borderline overlaps, the system’s use of an IoU-based threshold prevents triggering occlusion status unless the overlap is strong. Additionally, we tested with both dynamic (resume state update of occluded) and static occluded and found the dynamic approach to be robust across cases. While very rare edge cases (e.g., overlapping occlusions by multiple moving objects with missed detections) may still pose difficulty, we found such scenarios to be both rare and not significantly impactful to overall benchmark performance.
>
> ---
>
> **Q**: *Occluded objs are exempt from association ...?*
>
> **A**:
> Upon review, we realize that the phrasing in our original draft "Occluded tracklets are excluded from association ..." in line 229 was misleading. What we intended to say is that occluded tracklets are not excluded from the matching process (which actually is a part of association); rather, they are exempted from deletion to keep them for potential next matches upon receiving new detections.
>
> To clarify the process:
>
> 1- Association: All tracklets, including active and occluded ones participate in two rounds of matching (high and low confidence detections). If an occluded tracklet finds a suitable detection (based on IoU gating), it is updated and occlusion flag is set to False. So the matching logic does not change and same matching is applied for all cases.
>
> 2- Occlusion Handling: Tracklets are checked for possible occlusion based on IoU-based overlap with other tracklets. If detected as occluded, flags are set and corrective operators (e.g., velocity damping, box enlargement) are applied.
>
> 3- State Update: For matched tracklets, states are updated using Kalman posteriors with new observations. For unmatched (including occluded) ones, prediction-only updates are applied. Occluded tracklets are not removed unless the occlusion duration exceeds a defined threshold (so this one is the timing we consider for them called $T_{occ}$ in paper) and everytime we consider them for match upon receiving detections.
> We appreciate the reviewer for pointing this out. We revised this in the manuscript (lines 227-232 revised)
>
> ---
>
> **Q**: *dataset ...?*
>
> **A**: Yes, it is freely accessed by users and further details are added in appendix about annotation (kindly refer to answer 1 in general response above). Upon acceptance, we will add direct links to the paper as well.

---

> ### Author Response · Authors · 2025-11-23
> **Reply to reviewer xSDA (2)**
>
> **Q**: *Do you output predicted boxes...? how do enlarged boxes affect FP/FN and HOTA...?*
>
> **A**:
> Yes, we also log and output the occluded objects as they are available in the GT files for the evaluations (although occluded). To answer your question about impacts of occlusion handing on evaluations metrics, we provide an analytic explanation of the effect of box enlargement for the reviewer, that we also faced it in numerical results.
>
> For a fixed IoU threshold $\alpha$, HOTA factorizes as
> $
> \mathrm{HOTA}(\alpha) = \sqrt{\mathrm{DetA}(\alpha)\.\mathrm{AssA}(\alpha)},
> \qquad
> \mathrm{DetA}(\alpha) = \frac{TP_\alpha}{TP_\alpha + FN_\alpha + FP_\alpha}.
> $
> If we slightly enlarge the predicted box of tracklet $t$ while keeping the same center, and the enlargement factor is $(1+\varepsilon)$ in width and height, then the IoU with the ground-truth box becomes
> $
> \mathrm{IoU} = \frac{1}{(1+\varepsilon)^2}.
> $
> Because $\varepsilon$ is small and $\alpha \le \tfrac{1}{(1+\varepsilon)^2}$ for the threshold used in HOTA, (almost) every detection of $t$ remains a true positive, so $(TP_\alpha,FN_\alpha,FP_\alpha)$ and the association pattern stay unchanged. Consequently, both $\mathrm{DetA}(\alpha)$ and $\mathrm{AssA}(\alpha)$ and thus $\mathrm{HOTA}(\alpha)$ are almost identical to the original values.
>
> In contrast, if we completely remove tracklet $t$ over $T$ frames ( the baseline case which loses the id completely), each of its detections turns from a true positive into a missed ground-truth object. At threshold $\alpha$, this gives
> $
> TP_{\alpha}' = TP_{\alpha} - T,\quad FN_{\alpha}' = FN_{\alpha} + T,\quad FP_{\alpha}' = FP_{\alpha},
> $
> so the denominator $D_\alpha = TP_\alpha + FN_\alpha + FP_\alpha$ is unchanged and
> $
> \mathrm{DetA}'(\alpha)
> = \frac{TP_\alpha - T}{D_\alpha}
> < \frac{TP_\alpha}{D_\alpha}
> = \mathrm{DetA}(\alpha).
> $
> At the same time, all association contributions of $t$ are removed, which strictly decreases $\mathrm{AssA}(\alpha)$. Therefore
> $
> \mathrm{HOTA}'(\alpha) = \sqrt{\mathrm{DetA}'(\alpha)\.\mathrm{AssA}'(\alpha)}
> < \sqrt{\mathrm{DetA}(\alpha)\.\mathrm{AssA}(\alpha)}
> = \mathrm{HOTA}(\alpha),
> $
> while the slightly enlarged but still matched scenario leaves $\mathrm{HOTA}(\alpha)$ unchanged.

---

### Official Review · Reviewer_k4Wq · 2025-11-01

**Soundness:** 3
**Presentation:** 3
**Contribution:** 2
**Rating:** 4
**Confidence:** 4

**Summary:**

This work proposes a new method for multi-object tracking. It focuses on vehicle tracking on complex traffic scenes. It proposes a mechanism to handle occlusions by moderating Kalman Filter and enlarging the candidate box. It also introduces a tracklet refinement strategy, which uses scene information to improve trajectory continuity and accuracy. A new traffic benchmark is collected as well. Experiments on public benchmarks and the new one demonstrate the effectiveness of the proposed method.

**Strengths:**

1. The idea of handing occlusion situations and utilizing scene information makes sense.
2. The implementation of the occlusion handling and the scene prior constraints is reasonable.
3. The collected dataset can be helpful for the community.
4. The proposed method is effective on public benchmarks and the new one.

**Weaknesses:**

1. Experiments on efficiency are lacking. The paper claims that the tracker is real-time, but related experiments, like running speed, computational burden, etc., are missing. These experiments are needed to support the claim.
2. Methodology contribution is limited. This work focuses on the Kalman Filter-based data association process, and improves the process from multiple aspects. However, these improvements are more like engineering optimization, instead of methodology contribution from my view. It has practical value, but the methodology contribution is not enough.

**Questions:**

How's the method's performance on BDD100k? BDD100k[1] is a popular MOT dataset with multiple classes on traffic scenes. It should be included for dataset and method comparison.

Ref.

[1] Yu, Fisher, Haofeng Chen, Xin Wang, Wenqi Xian, Yingying Chen, Fangchen Liu, Vashisht Madhavan, and Trevor Darrell. "Bdd100k: A diverse driving dataset for heterogeneous multitask learning." CVPR 2020.

---

> ### Author Response · Authors · 2025-11-23
> **Reply to reviewer k4Wq**
>
> **Q**: *BDD100K*
>
> **A**:
> We thank the reviewer for the suggestion to include BDD100K benchmarks. We conduct a round of evaluation for FastTracker on BDD100K. The benchmarks results are reported directly from their original papers, assuming they are optimized in HPs perspective. We hope this addition addresses the reviewer’s concern and we also would like to add that the reason why these exps/results were not included in the main submission was only due to space limitations of ICLR and the already extensive set of original experiments we included (covering five major benchmarks MOT16, MOT17, MOT20, DanceTrack, and FastTrack):
>
> | Method              | mMOTA ($\uparrow$) | mIDF1 ($\uparrow$) |
> |---------------------|--------------------|---------------------|
> | SORT                | 30.9               | 41.3                |
> | DeepSORT            | 24.5               | 38.2                |
> | MOTDT               | 26.7               | 39.8                |
> | BYTE (with ReID) | 40.1               | 52.8                |
> | QDTrack             | 36.6               | 50.8                |
> | TETer               | 39.1               | 53.3                |
> | Unicorn             | 41.2               | 54.0                |
> | MOTR                | 35.5               | 48.2                |
> | FastTracker         | **43.8**           | **56.2**            |
>
> These are added to the revised draft (appendix F.4).
>
> For a simple run, please follow with this command in our added repo:
> ```
> python tools/bdd100.py bdd100k -f exps/example/mot/yolox_x_mix_det.py -c pretrained/yolox_x_mot17.pth.tar --fp16 --fuse --save_result
> ```
>
>
> ---
> **Q**: *Contributions ...*
>
> **A**:
> We detailed an answer to this concern in part 2 above in general response, please kindly refer to that comment for additional details. To summarize, as noted in 2 above, many existing works focus on heavier architectures or joint models to improve re-ID quality, or generally saying, more expressive re-IDs. While we respect that direction, our priority has been practicality delivering high performance with low overhead and enabling deployment across a wide range of platforms, including edge devices (now supported by our Jetson results in complexity analysis comment above and we will plan to test same for a mobile phone device). Rather than increasing complexity, we focused on strengthening lightweight parts of the pipeline. We hope this clarifies one of other important aspects of our contributions for the reviewer.
>
> ---
> **Q**: *running speed, computational burden, etc., are missing ...*
>
> **A**: Now added in 'RunTime-Complexity' response above. Besides, they are added to the revised draft (all added in appendix E).

---

### Author Response · Authors · 2025-11-23
**General response**

We would like to thank all reviewers for their valuable feedback and comments. Below we provide a brief general answer that touches on the main questions and requests raised (more details separately provided later in each reviewer's response).

---

1- Regarding dataset release and code repository (reviewers **Rd4h** and **xSDA** and **X9FY**):
Our code and data are available to users under the MIT license and the BigScience OpenRAILS-M license, respectively. They can be accessed and used freely without restrictions. However, to preserve anonymity for the review process, we did not include the direct links in the paper. Instead, we have prepared an anonymous GitHub in supplementary materials
containing an easy-to-follow readme instructions. In addition to python/torch, we provided C++ deployment in our repo, making deployment on edge devices such as Jetson and mobile platforms using NCNN feasible.
The annotations in our benchmark follow the standard MOTChallenge format, so users only need minimal adjustments. The only difference is that MOTChallenge includes only pedestrian labels, but our benchmark has multiple classes. Our dataset currently contains 9 training videos, 1 validation video, and 2 test videos (details added in appendix).
For future steps, we plan to deploy an evaluation server that researchers can upload their results and automatically receive evaluation metrics online.

---

2- Regarding the reviewer’s comment on the methodological contribution and use of hand-crafted designs, we would like to clarify:

- (For reviewer **X9FY**): We put most efforts in designing each component of our approach to effectively use environmental and contextual info, while keeping it computationally cheap. It is consistent with recent works like SparseTrack and PD-SORT. While our method has handcrafted thresholds and region-based logic, we cared to ensure robustness by validating across many scenarios (12 different envs) and tuning thresholds systematically to generalize well across datasets with a confidence threshold. Thus the sensitivity to many envs (e.g. crosswalks, roads, etc.) is tested with a single set of HPs.

- (For reviewer **k4Wq**): In terms of architectural innovation, one common direction in the field is to enhance tracking performance through deep re-ID modules. However, our focus is on building a lightweight, accurate, and deployable alternative that operates reliably even on edge devices (it is supported now by our new deployment results on Jetson device). Almost all compared benchmarks in paper rely heavily on computationally expensive re-ID modules, a few of them performs real-time capability on Jetson-class or even desktop-level GPUs.
We believe that enabling practical and real-world deployment of an academic methods is a meaningful direction.
One of main aspects of our contributions is applicability of our academic invention into broad applications of industry (covering strong GPUs, edge devices such as Jetson, and now even mobile phones because of deployed c++ version although not tested and planned to be included later) and we hope that the reviewer consider this as well. Besides, we provided a full table of results on BDD100K dataset as reviewer k4Wq requested.

---

3- (For reviewers **X9FY** and **k4Wq**): We report the runtime complexity for all YOLOX variants corresponding to the results presented in the main paper. Not only for a desktop level RTX 4060 GPU, but also on the edge device Jetson AGX Orin to support our claim about inference in edges devices. We post it as a separated comment.


---

4- About deviation and errors resulting from imperfect RoI (reviewer **X9FY** and **Rd4h**):

Our proposed hyperparameters for ROI and direction-angle, which are environment-aware, were tested across all scenarios in the FastTracker benchmark and showed consistent performance. The system is robust to noise and inaccuracies in ROI selection. For example, small deviations in $\theta$ do not impact results due to our confidence threshold as already tested. However, we acknowledge that severe misconfigurations (e.g., a 180 degrees reversal of road direction) can degrade performance. So in practical setups, such significant errors are unlikely, and overall the method shows good tolerance to real-world imperfections. In case of not-registered RoI, the tracking skips using environmental aware info and goes with occlusion handling and del/init policy (which is done for the MOTs and dancetrack benchmarks).

---

5- Reviewer **xSDA** suggested some clarifications (e.g. CP(t,t'), association and matching steps of occluded obj, etc.), all of which we have incorporated and reflected in the revised manuscript.

---

6- Reviewer **Rd4h** requested details about the motion model and the Kalman filter along with the noise parameters, and we have fully added these explanations in the appendix.

---

> ### Author Response · Authors · 2025-11-23
> **RunTime-Complexity**
>
> Here are our results for the YOLOX variant timing and GFLOPs, using input size 1088$\times$608 and implemented in torch with a desktop level RTX 4060 GPU. Based on references [1], [2], and [3], which state that FPS above 24 is considered real-time, all of our variants operate in real-time.
>
> | Model       | Detection Time (ms) | Final Time (ms) | Total FPS ($\uparrow$) | GFLOPs |
> |-------------|----------------------|------------------|----------------|--------|
> | YOLOX-X     | 35.7                 | 40.0             | 25.0       | 455.3  |
> | YOLOX-L     | 33.1                 | 37.4             | 26.7       | 251.3  |
> | YOLOX-M     | 30.9                 | 35.6             | 28.4       | 118.7  |
> | YOLOX-S     | 28.5                 | 32.8             | 30.5       | 43.0   |
> | YOLOX-Tiny  | 28.1                 | 32.5             | 30.9       | 24.5   |
> | YOLOX-Nano  | 25.8                 | 30.2             | 33.2       | 4.0    |
>
> According to the tracking complexity and timing, averaging on the tested benchmarks, ByteTrack runs at around 4.1 ms per frame (∼244 FPS), while our tracker runs at 4.3-4.4 ms (∼233 FPS). The small increase comes from adding occlusion detection, box enlargement and velocity damping for better re-identification, and conducted track initialization and deletion policies.
>
> In addition to the PyTorch implementation, we provide a full C++ deployment of our system, available in our repo. The users easily can deploy our detection and tracking pipeline on NVIDIA edge devices, such as the Jetson Orin. Below, we report the performance of our pipeline on the Jetson AGX Orin, based on our tests using both Torch and TensorRT (C++) implementations.
>
> | Model       | Jetson Orin (Torch) | Jetson Orin (TensorRT C++) |
> |-------------|----------------------|------------------------------|
> | YOLOX-X     | ~13.0 FPS            | ~19.5 FPS                    |
> | YOLOX-L     | ~14.0 FPS            | ~21.0 FPS                    |
> | YOLOX-M     | ~15.0 FPS            | ~22.5 FPS                    |
> | YOLOX-S     | ~16.3 FPS            | ~24.5 FPS                    |
> | YOLOX-Tiny  | ~16.5 FPS            | ~24.8 FPS                    |
> | YOLOX-Nano  | ~18.0 FPS            | ~27.0 FPS                    |
>
>
> [1] A. Plyer, G. Le Besnerais, and F. Champagnat, “Massively parallel Lucas
> Kanade optical flow for real-time video processing applications,” Journal
> of Real-Time Image Processing, pp. 713–730, 2016.
>
> [2] S. Wan, S. Ding, and C. Chen, “Edge computing enabled video
> segmentation for real-time traffic monitoring in internet of vehicles,”
> Pattern Recognition, vol. 121, p. 108416, 2022.
>
> [3] G. Ananthanarayanan, P. Bahl, P. Bod´ık, K. Chintalapudi, M. Philipose,
> L. Ravindranath, and S. Sinha, “Real-time video analytics: The killer
> app for edge computing,” Computer, vol. 50, no. 10, pp. 58–67, 2017.

---

### Author Response · Authors · 2025-12-02
**Evaluation of paper after Openreview accident**

We would like to thank the reviewers again for their time and constructive comments.
We made every effort, specially during rebuttal time, to address all raised concerns thoroughly. Many of the reviewers’ key questions (e.g., dataset release details, computational complexity, edge-device performance) were clarified in the response, and additional clarifications were provided where requested.
Because **all of our reviewers submitted borderline scores**, we really hoped that our detailed clarifications and new results would be considered for score updates.

However, given the unexpected situation on OpenReview, honestly, it gets very disappointing for us to miss such chance.

We therefore, respectfully ask the **new AC** to closely examine the correspondence between each reviewer concern and the content of our rebuttal. We believe the technical issues raised are fully addressed. We, of course, are open to reply back if there is any unclear part is left for the new AC as well.

Considering all of these, we hope the AC can understand the authors perspective and comes up to a fair decision at the end.
Bests regards,
Authors

---

### Meta-Review · Area_Chair_4Vie · 2026-01-04

**Summary:**

This paper presents FastTracker, a lightweight multi-object tracking system for vehicle-centric traffic scenes, incorporating occlusion-handling heuristics, scene priors, and a new benchmark dataset. Reviewers find the approach practically motivated and empirically solid, and the rebuttal clarifies implementation details and adds supporting experiments. However, the work primarily builds on existing frameworks with heuristic refinements, and the methodological novelty and conceptual depth remain limited, with concerns about reliance on manually defined priors and generalization. Overall, these issues outweigh the incremental empirical gains.

**Reviewer Concerns:**

**Concerns largely addressed by the rebuttal:**

- The rebuttal clarifies the runtime efficiency and occlusion handling through additional experiments and technical details.

- Additional evaluations (e.g., BDD100K) further support the empirical effectiveness of the approach.

**Concerns that remain outstanding:**

- Methodology contribution is limited.

- Reliance on manually defined scene priors and thresholds limits generalization and weakens the claimed real-time and deployment advantages as principled contributions.

**Reviewer Scores:**

The reviewers may remain cautious and are likely to maintain their original scores.

---

### Decision · Program_Chairs · 2026-01-26

Reject